



# Implementation and sensitivity analysis of a Dam-Reservoir OPeration model (DROP v1.0) over Spain

Malak Sadki[1], Simon Munier[1], Aaron Boone[1], and Sophie Ricci[2]

[1]CNRM, Université de Toulouse, Météo-France, CNRS, Toulouse, France
[2]CECI, CERFACS/UMR5318 CNRS, Toulouse, France

**Correspondence:** M. Sadki (malak.sadki@meteo.fr)

**Abstract.** The prediction of water resource evolution is considered to be a major challenge for the coming century, particularly in the context of climate change and increasing demographic pressure. Water resources are directly linked to the continental water cycle and the main processes modulating changes can be represented by global hydrological models. However, anthropogenic impacts on water resources, and in particular the effects of dams-reservoirs on river flows, are still poorly known and

generally neglected in coupled land surface - river routing models. This paper presents a parameterized reservoir model, DROP (Dam-Reservoir OPeration model), based on Hanasaki's scheme to compute monthly releases given inflows, water demands and the management purpose. With its significantly anthropized river basins, Spain has been chosen as a study case for which simulated outflows and water storage variations are evaluated against in situ observations over the period 1979-2014. Using a default configuration of the reservoir model, results reveal its positive contribution in representing the seasonal cycle of dis-

charge and storage variation, specifically for large-storage capacity irrigation reservoirs. Based on a bounded version of the Nash-Sutcliffe Efficiency (NSE) index, called $C_{2M}$, the overall outflow representation is improved by 43% in the median. For irrigation reservoirs, the improvement rate reaches 80%. A comprehensive sensitivity analysis of DROP model parameters was conducted based on the performance of $C_{2M}$ on outflows and volumes using the Sobol method. The results show that the most influential parameter is the threshold coefficient describing the demand-controlled release level. The analysis also reveals the

parameters that need to be focused on in order to improve river flow or reservoir water storage modeling by highlighting the difference in the individual effects of the parameters and their interactions depending on whether one focuses on outflows or volume mean seasonal patterns. The results of this generic reservoir scheme show promise for modeling present and future reservoir impacts on the continental hydrology within global land surface - river routing models.

Keywords : *reservoir modeling, large-scale hydrological modeling, sensitivity analysis, Sobol method.*

# 1   Introduction

Dams are used to provide essential services to mankind in terms of economic, environmental, and social impacts. They provide water supply for domestic, industrial and irrigation needs, enable hydroelectric power generation and river navigation and prevent extreme hydrological events. There are currently more than 58,700 large dams (heights > 15 m) worldwide, with an estimated cumulative storage capacity between 7,700 km³ and 8,300 km³ (Vörösmarty et al., 2003; Downing et al., 2006;





Lehner et al., 2011; ICOLD, 2020). When including millions of smaller dams ($\sim$ 16.7 M larger than 0.01 ha (Lehner et al., 2011)), the total global impounded water may even exceed 10,000 km$^3$ (Chao et al., 2008). This means that reservoirs hold more than four times the amount of water stored in rivers (the average annual river water storage ranges from approximately 1,200 to 2,120 km$^3$), and they account for approximately 20% of average annual river flow to the oceans (40,000 - 45,500 km$^3$/year) (Baumgartner and Reichel, 1975; Oki and Kanae, 2006; Syed et al., 2010; Lehner et al., 2011). More than 60% of the world's

largest rivers are fragmented by the construction of dams, which account for 90% of the flow from these river basins (Dynesius and Nilsson, 1994; Revenga et al., 2005; Grill et al., 2015, 2019).

Several studies have demonstrated the significant impact of reservoirs on river flow regimes at not only local scales, but also at larger regional and global scales: reservoirs impact the magnitude of downstream river flows and alter the temporal pattern of river discharge over the continental surface (Haddeland et al., 2006; Hanasaki et al., 2006; Döll et al., 2009; Biemans et al.,

2011; Shin et al., 2019; Gutenson et al., 2020). Through surface evaporation and water exchanges with groundwater, lakes and floodplains, reservoirs not only affect the water budget over land but also throughout the earth's horological cycle by interacting with the atmosphere and oceans (Pokhrel et al., 2012; Zhao et al., 2012; Wada et al., 2016; Frederikse et al., 2020). There is therefore an increasingly pressing need to integrate reservoir operations in large-scale land surface and global hydrological models (LSMs-GHMs) to overcome the existing biases in continental water cycle and river flow modeling given the number

of highly regulated basins.

Models developed to date which represent reservoir releases at a large scale can be categorized into data-driven and process-based approaches. The first category of models are built on the basis of observed release data, water levels and volumes. These methods range from simplified representations of reservoir operation using linear or multilinear regression (e.g., Young Jr, 1967; Raman and Chandramouli, 1996), to very sophisticated models based on machine learning and artificial intelligence

techniques, such as neural-network-based methods (e.g., Maier and Dandy, 2000; Razavi and Karamouz, 2007; Ehsani et al., 2016; Coerver et al., 2018). However, this approach requires specific knowledge of the studied reservoirs and requires access to a large amount of observed data. This approach also remains limited in its applicability for future predictions since the generated operational rules are based on historical data and thus do not take into account future potential social economic and predicted climatic changes in the operation of these reservoirs. Process-based approaches, on the other hand, are based

on conceptualizing reservoir responses according to its operational purpose by linking release control to physical processes, such as crop growth and the associated water requirements, or to water and energy demands that can be estimated at the global scale. The representation of dam operations is thus achieved without having to explicitly observe the actual release operations performed on each reservoir (Gutenson et al., 2020). The best known schemes in this category are those developed by Hanasaki et al. (2006) and Haddeland et al. (2006), which are inflow-and-demand-based models as presented by Yassin et al.

(2019). These two generic models have been implemented in several global hydrological and water management models (e.g., WaterGAP (Döll et al., 2009)), VIC (Haddeland et al., 2006), H08 (Hanasaki et al., 2008), and PCR-GLOBWB (Van Beek et al., 2011)).

Of all the studies which have been carried out with these models, very few have been focused on Iberian Peninsula basins where the prevailing semi-arid climate leads to a greater necessity to store water in large-capacity reservoirs which lead to



larger reservoir effects on rivers (Batalla et al., 2004; López-Moreno et al., 2009; Lorenzo-Lacruz et al., 2010). Spain is, in fact, among the top ten dam-building countries with more than 1,064 dams (*ICOLD, 2020*). The study of Grill et al. (2019), which assessed global river connectivity, revealed that the regulation effect of dams is the dominant pressure source in Spain's rivers, where very high degrees of regulation cause an alteration of the natural river flow regime for its five main rivers.

This study proposes a global and parameterized reservoir model, DROP, built on the basis of the generic scheme by Hanasaki
et al. (2006) and the first aim of the study is to implement it in Spain as a case study to represent reservoir releases. The second aim is to provide a comprehensive understanding of how uncertainties in each of the model parameters are affecting the overall accuracy of its predictions. In fact, apart from sensitivity and parameter tuning tests in Hanasaki et al. (2006) and Shin et al. (2019), the authors are unaware of an exhaustive sensitivity analysis conducted on this model to date. Further work will focus on implementing this scheme in global hydrological models in order to provide a physical representation of reservoirs on large
scales.

This paper is organized as follows: Section 2 provides a description of the DROP model and a theoretical outline of the sensitivity analysis method. The study area in Spain, the available observational data, the model setup and the sensitivity analysis implementation are described in section 3. Sections 4-5 illustrate and discuss the model's overall performance and the parameter sensitivity results. Conclusions and perspectives of the study are presented in the last section.

## 2 Methodology

### 2.1 DROP: a global Dam-Reservoir OPeration model

The parameterized DROP model has been developed based on Hanasaki et al. (2006) scheme. The model works at the level of each reservoir individually and is based on the mass balance (as shown below in Equation 1) of each reservoir to calculate the release at its outlet.

$$\frac{dV}{dt} = Q_{in} - Q_{out} \qquad (1)$$

$Q_{in}$ and $Q_{out}$ stand respectively for the net inflow to the reservoir and its outflow at the outlet. In fact, the net inflow, $Q_{in}$, combines different physical processes, as shown in Equation 2: it includes water inputs from precipitation $P$, direct runoff $R_d$ and tributary inflows $Q_{trib}$, but also accounts for evaporation losses $E$ and ground water exchanges $Q_{gw}$.

$$Q_{in} = (P - E) \times A_{reservoir} + Q_{trib} + R_d \pm Q_{gw} \qquad (2)$$

where $A_{reservoir}$ is the reservoir surface area.

In order to simulate dam releases, $Q_{out}$, the reservoir model categorizes reservoirs into irrigation and non-irrigation reservoirs, computes the mass balance for each reservoir individually and calculates releases based on inflow and water demands. A schematic of the DROP model is shown in Figure 1 with its 6 parameters (in blue). To give an overview of its overall functioning, the model takes as input inflow and water demands and calculates at a monthly time step dam releases according to
the reservoir management purpose and its relative capacity compared to inflow, noted $c$. Operating rules are set following an





'operational year'. This type of year has been introduced by Hanasaki et al. (2006) and it differs from hydrological and calendar years. It starts the first month of the water release period and is therefore specific to each reservoir. This specific month, representing the $1^{rst}$ parameter of the model, will be noted as $m_{start}$ in the remaining sections of this article. At the beginning of each operational year, defined by $m_{start}$, reservoir storage volume $S_{init}$ is compared to the ideal filling value $S_{ideal}$, defined defined as a ratio, $\alpha$, of storage capacity $C$. Reservoir simulated releases are impacted by this step as they are retrospectively revised upwards or downwards depending on whether the reservoir has more or less water storage than the ideal rate. Dam monthly releases are then computed following two steps: first, a provisional release is calculated based on water demands and the annual mean inflow. In irrigation reservoirs, two parameters are involved: $d_{max}$, setting the control area of the reservoir and thus the water needs to be supplied, and $M$, defining the minimum release to be provided for environmental requirements. The calculation scheme remains simplistic for other management purposes where releases are set to mean annual inflow. The provisional release is then corrected by incorporating a "demand-controlled release" ratio $R$, controlled by two parameters $c_{threshold}$ and $b$, which accounts for inflow pattern influence in reservoirs with low storage capacities compared to inflow. The DROP model therefore counts 6 parameters, as shown in Figure 1.

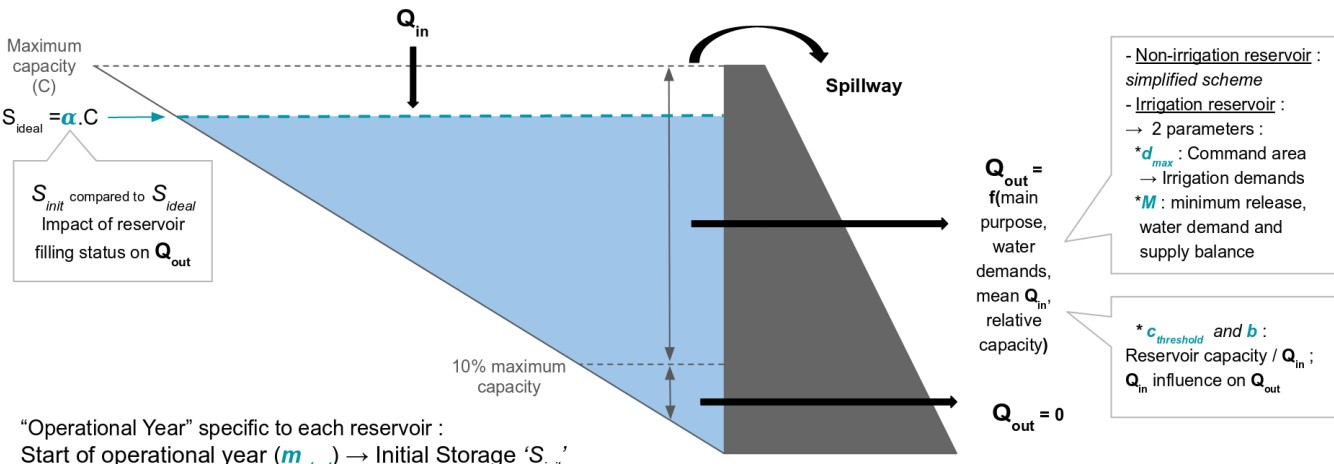

**Figure 1.** Schematic representation of the DROP model showing its 6 parameters (in blue): $m_{start}$ defines the start of reservoir operational year. $\alpha$ sets the ideal filling rate of the reservoir to be reached at each starting month of the year. The remaining 4 parameters are used in dam release computation: $d_{max}$ and $M$, operating only in irrigation reservoirs, set respectively the control area of the reservoir (and therefore irrigation water demands) and minimum release to be provided. $c_{threshold}$ and $b$, on the other hand, account for inflow pattern influence in reservoirs with low relative storage capacity to inflow (i.e. run-of-river reservoirs).

The operating rules are detailed below. First, at the beginning of each operational year, an annual release coefficient $K_y$ is computed to determine the filling rate of the reservoir. $K_y$ is a ratio between the reservoir storage at the beginning of the year $S_{first,y}$ and the long-term target storage $S_{ideal}$ :

$$K_y = S_{first,y}/S_{ideal}(\alpha C) \tag{3}$$





where $S_{ideal} = \alpha \times C$, $\alpha$ being a non-dimensional constant, and $C$ is the total storage capacity of the reservoir. $S_{ideal}$ represents the ideal filling level at the beginning of each year; $\alpha$ was set semi-empirically to 0.85 of the maximum capacity for all

reservoirs based on sensitivity tests conducted in Hanasaki et al. (2006). This annual release coefficient is the one used to weight the provisional releases calculated afterwards depending on whether the reservoir has more (Ky>1) or less (Ky<1) water storage than the ideal rate. The underlying aim is to reduce interannual variation in streamflow.

 The following steps describe how reservoir releases are computed. The provisional monthly releases, $r'_m$, are set depending on the reservoir's main purpose. The scheme is simplified for a non-irrigation reservoir since it constantly releases the mean

annual inflow $i_{mean}$ calculated over the whole simulation period (noted "long-term" in the remaining sections of this article):

$$r'_m = i_{mean} \tag{4}$$

For an irrigation reservoir, monthly releases are proportional to water demands. They are here parameterized following Shin et al. (2019) as:

$$r'_m = \begin{cases} i_{mean} \left[ (M + (1 - M)\dfrac{d_m}{d_{mean,y}} \right] & \text{if DPI} > 1 - M \\[2ex] i_{mean} + d_m - d_{mean,y} & \text{if DPI} \leqslant 1 - M \end{cases} \tag{5}$$

where $d_m$ and $d_{mean,y}$ are, respectively, monthly and annual mean water demands. DPI (Demand Per Inflow) is, as introduced by Shin et al. (2019), the ratio between $d_{mean,y}$ and $i_{mean}$, and $M$ represents the minimum monthly release as a percentage of $i_{mean}$. It is set to 0.5 by Hanasaki et al. (2006) and Döll et al. (2009), and to 0.1 by Biemans et al. (2011) and Shin et al. (2019).

 When the DPI is above the set threshold $(1 - M)$, water needs are considered to be very high and can only be partially

fulfilled by the water stored throughout the coming year. The priority is to first ensure a minimum release, $M \times (i_{mean})$, in order to meet the environmental flow requirements. The remaining part of the annual inflow is released throughout the year on a monthly basis following the sub-annual water demand fluctuation curve. Otherwise, when the DPI is below $(1 - M)$, the reservoir releases all the monthly water demand that is needed.

 In this scheme, only irrigation demand is considered. Since dams provide water for the downstream demand within a certain

distance, a maximum distance, $d_{max}$, is a parameter to define for each reservoir, the irrigated grid cells within the river basin to be supplied and thus delimits a "command area" to each reservoir. In contrast to Hanasaki et al. (2006) (where a crop growth model is considered to calculate the irrigation demand), here the distribution of irrigated areas is based on ECOCLIMAP SG (Calvet and Champeaux, 2020; Druel et al., 2021) which is used by the irrigation module in the ISBA LSM (Druel et al., 2021) to compute irrigation demands for 5×5 km resolution grid cells (see section 3). The irrigation water demands are

aggregated within the command area for each reservoir. Unlike Hanasaki et al. (2006) scheme, the DROP model entirely separates irrigation from non-irrigation reservoirs and only accounts for irrigation water demands for this category of reservoirs. Moreover, industrial and domestic demands are highly uncertain and hardly accessible. In fact, these consumptions are strongly linked to each country's specific political and economical policies and extensive records are generally not made public. Some global databases, such as the Food and Agriculture Organization of the United Nations (FAO) global information system





"AQUASTAT" AQUASTAT (1994), provide estimates of average annual water withdrawals by activity sector and by water resource publicly accessed at the country level. However, such estimates are still limited in terms of temporal and spatial resolution and would consequently add a non-negligible source of error to the model. They are therefore neglected in this study.

In all possible cases, regardless of the reservoir purpose, the water released over the operational year is equivalent to the

long-term mean annual inflow.

The release computed so far is provisional. The real monthly release is calculated as follows :

$$
r_m =
\begin{cases}
K_y \times r'_m & \text{if } c \geqslant c_{threshold} \\
(1 - R) \times i_m + R \times K_y \times r'_m & \text{if } 0 \leqslant c < c_{threshold}
\end{cases}
\tag{6}
$$

where $c$ is the relative capacity of a reservoir and is defined as the ratio between storage capacity $C$ and the long-term mean annual inflow water volume ($c = C/I_{mean}$). The parameter $R$, as introduced by Shin et al. (2019) (and also parameterized as

$\beta$ by Horan et al. (2021)), is a so-called "demand-controlled release ratio". It is parameterized in the current study as :

$$
R = \left( \frac{c}{c_{threshold}} \right)^b
\tag{7}
$$

where $R$ describes the influence of the inflow regime on release for small storage capacity reservoirs. It varies between 0 and 1 and includes two parameters: $c_{threshold}$ and $b$ coefficient. In fact, the smaller the reservoir capacity is compared to inflow, the closer it gets to run-of-the-river dams where release is close to the natural river flow and thus the influence of the inflow annual

pattern. Otherwise, when $c$ is above the threshold (large capacity dams), then $R = 1$ and the release is fully controlled by water demand. $c_{threshold}$ and $b$ are set, respectively, as 0.5 and 2 in both Hanasaki et al. (2006) and Biemans et al. (2011). Shin et al. (2019), on the other hand, proposed an analytical formula to compute $R$, which is reproduced in this study, and set $c_{threshold}$ and $b$ respectively to $1/\alpha$ and 1. The monthly releases are also weighted by the annual coefficient, $K_y$ from Equation 3, which is calculated at the beginning of the operational year and describes how full the reservoir is relative to the ideal rate. Reservoir

releases can therefore be revised upwards or downwards depending on whether the reservoir had more or less water storage than the ideal rate.

A description of the reservoir model parameters is summarized in Table 1.

The reservoir volume is derived at each time step from the water balance. Boundary conditions are defined considering two possible scenarios: i) if the reservoir is full, the excess water is spilled. ii) When reservoir storage falls below 10% of the

capacity, the reservoir reaches the dead storage zone and water release is prevented.

In order to run the model, reservoir characteristics, such as the storage capacity and main purpose, are needed. The model also requires continuous time series of inflow and water demands to compute releases. In the current study, all of the modeled reservoirs are located in Spain where both the physical characteristics, in-situ observations of natural and anthropized flows, and storage volumes are publicly available. Section 3 describes the global and local data sets used herein.





**Table 1.** Description of DROP model parameters

| Parameter | Description |
| --- | --- |
| $\alpha$ | Set ideal filling rate of reservoir |
| $d_{max}$ (km) | Set control area of reservoir |
| $m_{start}$ | Refer to first month of operational year |
| $M$ | Define minimum release to be provided for environmental requirements |
| $c_{threshold}$ | Threshold of relative capacity ; Influence of inflow pattern in reservoir release |
| $b$ | Influence of inflow pattern in reservoir release |

## 2.2   Sensitivity Analysis : *Sobol*'s method

The Sobol sensitivity analysis method (Sobol, 1993) has been widely used in hydrological models in order to identify the parameters that contribute the most to the model output uncertainty. It is a global variance-based approach where, for a chosen variable of interest $Y$, the total variance is decomposed into fractions attributed to each individual input as well as the interactions between them. If $Y = f(X)$ is a goodness-of-fit metric of the model with $X$ representing the set of parameters

$X_1, X_2, ... X_p$ of size $p$, the total variance of $Y$, $D(Y)$, is decomposed as follows (Zhang et al. (2013)):

$$D(Y) = \sum_i D_i + \sum_{i<j} D_{ij} + \sum_{i<j<k} D_{ijk} + ... + D_{12..p} \tag{8}$$

where $D_i$ is the amount of variance due to a parameter $X_i$ alone, $D_{ij}$ is the amount of variance arising from the interaction between the parameters $X_i$,$X_j$ etc. Equation 8 is known as Hoeffding decomposition (Hoeffding, 1948).

The sensitivity indices, called *Sobol's* indices, are computed as ratios between the component variances and total variance

in order to measure the contribution of each single parameter and each parameter interaction. The first-order *Sobol* index, $S1$, captures the sensitivity of $Y$ to each input parameter $X_i$ taken alone. The second, $S2$, and higher order indices describe the contribution of the multiple interactions between parameters. A model with $p$ parameters requires $2^p - 1$ indices to be evaluated, which rapidly becomes computationally challenging for high values of $p$. The total order index, $ST$, measures the full influence of a parameter by including all of the variance caused by its interactions with the rest of parameters, all orders

included. The first , second and total order *Sobol's* sensitivity measure formulas are:

$$S1_i = \frac{D_i}{D} \tag{9}$$

$$S2_{ij} = \frac{D_{ij}}{D} \tag{10}$$

$$ST_i = 1 - \frac{D_{\sim i}}{D} \tag{11}$$

where $D_{\sim i}$ is the amount of variance due to all of the parameters except for $X_i$.





Model parameter sampling and *Sobol* index estimation are here performed using the open-source Python library *SALib* (Herman, 2017). The parameter samples are generated following Quasi-Random sequences (Saltelli, 2002) in order to scatter the sample points as uniformly as possible over the parameter space. Following the theorem, the different orders of Sobol indices are estimated from a total number of model runs of $N \times (2 \times d + 2)$, where $N$ and $d$ are respectively the sample size and the number of parameters. The package also provides confidence intervals of the index estimates at the 95% confidence
level.

## 3    Study area and Data

### 3.1    Spain river basins description

Spain has an estimated area of 505,983 km$^2$ (INE, 2022), representing more than 85% of the Iberian Peninsula (estimated area 583,254 km$^2$, Lorenzo-Lacruz et al., 2012). It has river basins with sizes ranging from a few km$^2$ to more than 80,000 km$^2$ (i.e.
the Ebro basin). The five largest river basins are the Ebro, flowing into the Mediterranean Sea, and the Duero, Tajo, Guardiana and Guadalquivir, flowing into the Atlantic Ocean. They also represent the river basins where the natural river flow pattern is the most significantly altered in the country, with high degrees of regulation resulting from extensive reservoir construction. These different rivers have a Mediterranean hydrological pattern, which is characterized by high flows during the wet season (i.e. autumn and winter) and low flows during its dry season (i.e. late spring, summer). This seasonality explains the strong
anthropization of the Spanish hydrographic basins, and, in particular, the construction of more than 1,200 dams mainly in the second half of the 20$^{th}$ century. They are essential for retaining enough water to meet irrigation and domestic water demands, as well as for hydropower production that represents ~13% of the Spanish electricity generation (IEA, 2022, in 2020,).

### 3.2    Data set

The data series required as inputs to the model and the ones used to validate the outputs are taken from Spain database.
Reservoirs characteristics are taken from the global GRanD database. Irrigation demands are simulated by the ISBA irrigation module. The needed input data are detailed below :

–   Local (Spain) Database: In situ observations of natural and anthropized flow and volume data are made publicly available by the Center for Hydrographic Studies of Spain (CEDEX, Ministry of Public Works and Ministry for Ecological Transition, Spain). The national database includes the location and daily time series of discharge for 1,119 gauge stations
215        and outflows from 347 reservoirs over the period 1900-2014.

–   Global Reservoir and Dam (GRanD) database: from which the general characteristics of dams are taken (Lehner et al., 2011). Version 1.3 published in 2019 includes 7,320 reservoirs and provides the geographical location of dams as well as attribute information such as the construction year, maximum storage capacity, surface area and the set of purposes of reservoirs. 263 of the reservoirs listed in GRanD are located in Spain. Most of them were built from 1955 to 2000,
220        reaching a total storage capacity of 56,480 hm$^3$ in 2016, which exceeds the mean annual streamflow of the 8 major





rivers of the Iberian Peninsula (55,850 hm$^3$/year) (Lorenzo-Lacruz et al., 2012). Irrigation and hydroelectricity are the most identified purposes of more than half the reservoirs, and water supply comes third. In this paper, all management purposes different from irrigation are grouped in the 'other purposes' category.

– Simulated irrigation demands : The irrigation water demands are simulated by the new irrigation scheme implemented in
the ISBA LSM (Druel et al., 2021). It uses the ECOCLIMAP-SG land cover classification to identify the areas within the grid cell that can be irrigated. Three main parameters are set in the model to control respectively the irrigation triggering (when the plant reaches the wilting point), the period of crop growth where irrigation is possible (between emergence and harvest) and the amount of water used for irrigation. In the default configuration, a predefined amount of water of 30 mm is set for each irrigation and a 7-day minimum return period is fixed between two irrigation operations. These parameters
can be user-defined for each vegetation type and in each grid cell. To generate irrigation demands over Spain, we used the 5 km resolution SAFRAN-based meteorological datasets for that country (Quintana-Seguí et al., 2017) that was available over the period 1979-2014 to force the ISBA land surface model. Set to the default configuration, the irrigation module within the LSM computed daily irrigation demands for each grid cell over that time period. At the input of the reservoir model, $d_{max}$ is set beforehand and delimits for each reservoir a command area made of a selection of grid cells within
the same basin at a lower altitude. The equivalent amount of irrigation water requested from a reservoir in a given month corresponds to the aggregated daily irrigation demands of all crop types within the retained grid cells.

## 3.3 Model Setup : Pre-processing steps

### 3.3.1 Cross-referencing Global and Local data sets

Out of the 263 reservoirs listed in GRanD, only 216 were kept after cross-referencing the two databases and for which both
the characteristics and time series of release and volume could be identified. In fact, two reservoirs were doubly identified in GRanD v1.3 (which IDs were 2882, 2844) because they were rebuilt and/or renamed; their most recent characteristics are those retained. The 45 remaining reservoirs, located mainly in the northwest and south of Spain, were not identified in the Spanish database since they were built after 2014.

The maximum storage capacity of the chosen reservoirs goes from 9.5 (the San Lorenzo Mongay dam, located on the Segre
river in Ebro basin) to 3,200 hm$^3$ (the La Serena dam on the Zujar river in Gardiana basin) with a mean of 236 hm$^3$ and standard deviation of 441 hm$^3$. Using the latitude and longitude information, the chosen reservoirs were located on the new 1/12° resolution river network derived from CTRIP river routing model (Munier and Decharme, 2021). Dam locations were adjusted so as to have comparable drainage surfaces between those given by the GRanD database and those estimated by CTRIP.
An initial analysis conducted on observed river flows upstream and downstream of the reservoirs has identified a common seasonal behavior among those with irrigation which is that the peak dam release is shifted in time from the natural inflow. This is due to the typical operating mode of these reservoirs, which are designed to retain water arriving upstream during winter (wet season) and release it during summer (dry season) to meet irrigation needs.





### 3.3.2 Reconstructing Inflow

Based on the water budget equation, the net inflow $Q_{in}$, used as input to the reservoir model, was reconstructed at the level of each reservoir from observed time series of dam releases and volumes provided by the Spanish database. In this way, all the components of the water balance that are not accounted for in the reservoir model are indirectly represented, namely: direct runoff and precipitation water inputs, evaporation and infiltration water losses, groundwater exchanges (gains or losses) and water diversions through channels (Equation 2).

In fact, the DROP model is aimed to be coupled to a series of models that can represent these different processes. In the ISBA-CTRIP land surface - river routing model for example (Decharme et al., 2019), evaporation is computed by FLake, a module representing energy balance in lakes (Le Moigne et al., 2016), direct runoff is derived from ISBA land surface model, and inflows from tributaries as well as groundwater exchanges are computed by CTRIP river routing model (Munier and Decharme, 2021).

For each of the selected reservoirs, the longest continuous common period of daily observed outflows and volumes were first determined. At this stage, only reservoirs with more than three-year time series were retained, which leaves 215 reservoirs to be simulated. The net inflows were then derived from outflows and volume variations at daily scale (Equation 1). The computed net inflows were then corrected by removing outliers in two steps : first, all peak flows were selected when their maximum value exceeded five times the long-term mean. Among the peak values, outliers are then identified when relative difference of

slopes before and after peak flow is less than 10%. Both thresholds were set empirically. This step differentiates the outliers from the hydraulic behaviour of a river in flood recession periods. The outlier is replaced by a linearly interpolated value. The length of the corrected time series goes from 3.5 to 34 years with a median of 23 years. The main purposes, simulation period lengths and the relative capacities of the 215 reservoirs simulated are show in Fig. 2.

We note a good distribution of management purposes and relative inflow capacities in the final selected reservoirs. Overall,

half the reservoirs are primarily used for irrigation, which is mainly due to the semi-arid climate of the Iberian Peninsula and the high needs of irrigation in the country. The rest of the reservoirs are allocated to hydropower generation, water supply and different other purposes with respective percentages of 29%, 16% and 5%, and are grouped in the non-irrigation reservoir category.

### 3.4 Sensitivity Analysis Implementation

A sensitivity analysis with respect to the 6 parameters was conducted on the performance of a Nash-Sutcliffe Efficiency (Nash and Sutcliffe, 1970, NSE,) bounded version, called $C_{2M}$ (Mathevet et al., 2006), on outflows using the Sobol method. In fact, the NSE values in some reservoirs were highly negative for some simulations, thus this metric wasn't suitable for a variance-based sensitivity analysis method like Sobol's. $C_{2M}$ is used instead as it is a normalized version of NSE that varies between -1 and 1 and where all negative values are bounded between 0 and -1. Parameter default values, bounds and distributions are

listed in Table 2. The parameter distributions were all considered uniform except for relative capacity for which the distribution is logarithmic to align with the observed pattern on the modelled reservoirs (Fig. 2).





**Figure 2.** Main characteristics of the chosen reservoirs : (a) Classification of their main purpose, (b) distribution of simulation period lengths, and (c) Histogram showing decimal log values of their relative capacity.





**Table 2.** Summary of the DROP model parameter default values and feasible ranges for the sensitivity analysis

| Parameter | Default value | Min value | Max value | Distribution |
|---|---|---|---|---|
| $\alpha$ | 0.85 | 0.6 | 0.95 | Uniform |
| $d_{max}$ (km) | 100 | 1 | 250 | Uniform |
| $m_{start}$ (irrigation;other) | 4;5 | 1 | 12 | Discrete Uniform |
| $M$ | 0.5 | 0 | 1 | Uniform |
| $c_{threshold}$ | 0.5 | 0.001 | 20 | Logarithmic |
| $b$ | 2 | 0.5 | 5 | Uniform |

Default values for $\alpha$, $M$, $c_{threshold}$ and $b$ are those considered in Hanasaki et al. (2006). $\alpha$ selected bounds cover a realistic range of ideal reservoir filling rates. The distribution and bounds values of $c_{threshold}$ parameter are drawn from the relative capacity distribution of the 215 modeled reservoirs (Fig. 2). $b$ lower limit is 0.5, below which the shift to the demand-controlled

state of a reservoir becomes too abrupt once $c$ exceeds $c_{threshold}$. The upper limit considered for $b$ is 5, beyond which the transition curve between the behavior of a low relative capacity reservoir and a high relative capacity one becomes unchanged, following sensitivity tests run conducted on this parameter. For $d_{max}$, both the default value and the lower and upper limits were set to be consistent with the size of Spain river basins. The operational year starting month, $m_{start}$, is set to April for irrigation reservoirs in order to match the beginning of crop irrigation season considered in the irrigation model. For other

reservoirs, $m_{start}$ is chosen empirically to be May as default value based on the observed filling curves of the reservoirs, which tend to be at the maximum filling level near May.

The sensitivity analysis was performed on each of the 215 reservoirs separately, distinguishing between irrigation and non-irrigation reservoirs since the number of parameters involved depends on the main purpose of the reservoir (6 and 4, respectively, as $d_{max}$ and $M$ are only considered in irrigation reservoirs).

Using Saltelli's quasi-random sampling method (Saltelli et al., 2010), a sample size of 4,096 was used for this analysis in each category, resulting in 4,096×(2×6+2)=57,344 and 4,096×(2×4+2)=40,960 model runs for each of the 107 irrigation and 108 non-irrigation reservoirs, respectively. By comparing with smaller sample sizes, the confidence intervals estimated by SALib show that the Sobol indices converged and that the chosen sample size is sufficient to reliably represent the results.

## 4 Results

This section presents the main results of this study. First, simulation results of the reservoir releases using the default configuration of the model are displayed. Then, a sensitivity analysis of the model parameters is presented.



## 4.1 Reproducing the flow seasonal shift in irrigation reservoirs

Using the default parameterization with the parameters listed in Table 2, an operating rule is determined for each of the 215 reservoirs, and both outflows and water storage variations are simulated through complete operational years within the overall period 1979-2014. Fig. 3 shows the $C_{2M}$ values at reservoir outlets by evaluating the simulated monthly outflow time series against in situ observations. The left panel shows the results obtained using a reference simulation where rivers are considered in their natural state ($Q_{out} = Q_{in}$). The right panel shows the $C_{2M}$ improvement rate when considering the DROP model.

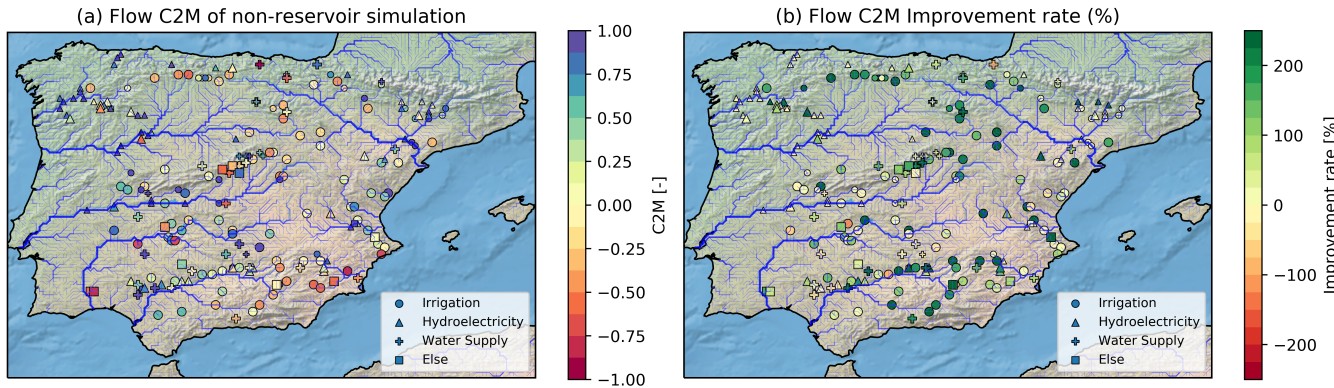

**Figure 3.** Results of the DROP model contribution (default configuration) in river flow modeling at reservoir outlets : (a) $C_{2M}$ performance metrics of non-reservoir simulation (reference simulation where $Q_{out} = Q_{in}$) and (b) $C_{2M}$ improvement rates by integrating the reservoir model. Symbols represent the different reservoir management purposes. They come in two different sizes: larger if c $\geqslant$ 0.5 and smaller if c<0.5.

Overall, with DROP, the river flow representation is clearly improved at nearly all reservoirs' locations. The median $C_{2M}$ index for flows is 0.52, which corresponds to a 43% improved flow representation when compared to natural river representation. For storage volumes representation, the mean correlation is 0.53 with a standard deviation of 0.3. Half of the reservoirs have a correlation greater than 0.63 between observed and simulated storage volumes.

The results reveal the model's positive contribution in representing the seasonal cycle of river flow, specifically for irrigation large-storage capacity reservoirs, as the model reproduces the seasonal shift between inflows and outflows caused by irrigation management rules with reasonable accuracy. For these reservoirs, the correlation between simulated and observed discharge increases from 0.49 (reference simulation) to 0.75 in the median. Regarding storage volumes, correlation reaches 0.74 in the median. As an example, the "Gonzalez Lacasa" reservoir located in the Ebro basin shows typical results in Fig. 4. In fact, this irrigation reservoir stores incoming water during winter / early spring, which explains a lower outflow than inflow (respectively in blue and orange for observed discharges, panels (a),(b) in Fig. 4) between October and March, and releases the water in summer when there is insufficient water to supply all crops' irrigation needs. The period of release, from April to September in this case (shown in the annual cycle, panel (b) in Fig. 4), corresponds to crops growing period and therefore





to high irrigation water needs. As a result, the discharge seasonal curve is shifted and the maximum monthly discharge, in this example, is reached in July instead of April. This management scheme is well reproduced by the DROP model, as the simulated outflows (shown in red) align well with the observations (in blue), and consequently so do the volume variations.

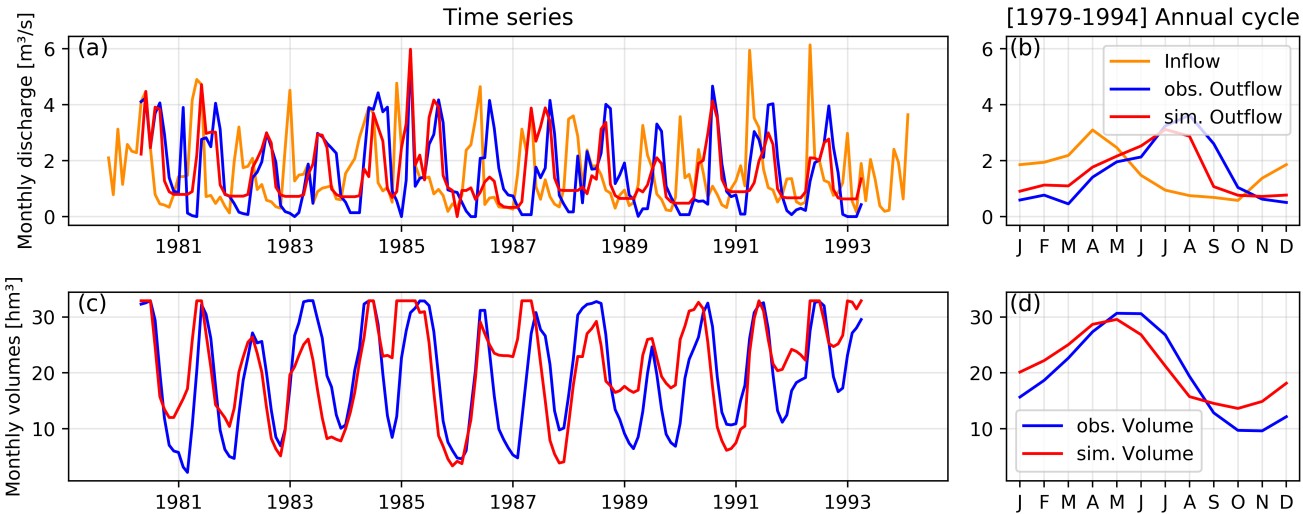

**Figure 4.** Simulation results for the 'Gonzalez Lacasa' irrigation reservoir within the Ebro river basin over the period [1979-1994]. The monthly time series of dam releases and storage volumes are shown in panels (a) and (c), respectively (observations are indicated in blue and simulations in red). Their corresponding mean annual cycles are shown in panels (b) and (d), respectively.

The improvement of the $C_{2M}$ distribution for each category of dams, in terms of main purpose and relative capacity, is
shown in Fig. 5. Note that for irrigation reservoirs, the improvement rate reaches 80% in terms of the median, and 123% for those considered of high relative capacity (here c >= 0.5) and which are fully demand-controlled. The flow improvement rates for the rest of the high relative capacity reservoirs are dispersed but remain positive at the median despite the simplistic approach of DROP. For low relative capacity reservoirs (here c < 0.5) and independent of the management purpose, the model's contribution is almost null since the reservoirs are considered "run-of-river" and the influence of inflow regime is predominant.

**4.2  Results of the sensitivity analysis**

The distribution of first-order Sobol indices for each parameter calculated at each of the 107 irrigation and the 108 non-irrigation reservoirs are shown in Fig. 6 with a box-plot. Overall, it emerges from Fig. 6 that the most influential parameter is $c_{threshold}$. In fact, based on the definition of $S1$, $\sim 48\%$ of the total variance in $C_{2M}$ is attributed to $c_{threshold}$ alone within irrigation reservoirs. In non-irrigation reservoirs, this parameter accounts for $\sim 74\%$ of $C_{2M}$ variance. The $M$ parameter is
ranked second in irrigation reservoirs and accounts for 15% alone in median of all of the variance, followed by $d_{max}$ with a S1 index of 0.03. The parameter controlling the month for which the operational year starts, $m_{start}$, alone has very little influence

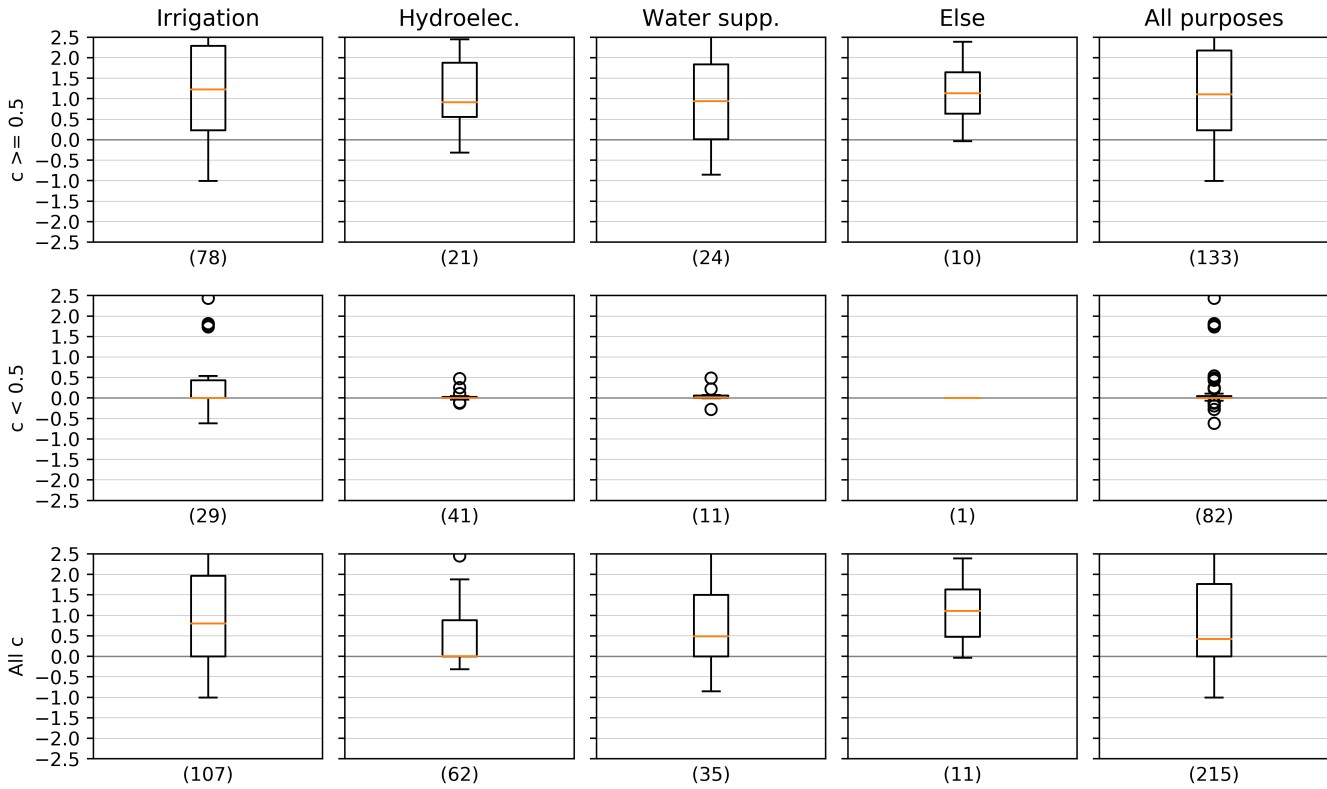

**Figure 5.** Distribution of river flow, $C_{2M}$, improvement rates with the DROP model based on their main purpose (columns) and relative capacity (rows). The x-axis labels show the number of reservoirs in each category, y-axis shows $C_{2M}$ improvement rates.

on the overall outflow $C_{2M}$ variance, although the effect is slightly more noticeable in the non-irrigation reservoirs. $\alpha$ and $b$ are considered the least influential parameters for most reservoirs.

To better illustrate how each parameter individually affects the model outputs, the reservoir *'Gonzalez Lacasa'* (GRanD ID
2699 ; $c$=0.65) is set as an example. Given the sensitivity analysis results, a screening step is added on $c_{threshold}$, $M$ and $\alpha$ separately and the rest of the parameters are set to their default values. The means of monthly outflows over the simulation period [1979-1994] are shown in Fig. 7.

Regarding the first parameter, and for values of $c_{threshold}$ below the relative capacity of the reservoir (lower than 0.65 for *'Gonzalez Lacasa'* case), the reservoir is considered with high storage capacity and so monthly releases are completely driven
by the demand, resulting in a seasonal shift between inflow and outflow and a peak of discharge in July in order to meet the irrigation needs. Conversely, when the $c_{threshold}$ is higher, the dam is considered to have low storage capacity which reduces the buffering effect and increasingly aligning the simulated release curve (in red) with the seasonal trend cycle of inflow (in orange), as shown in Fig. 7 (a). The buffer role of the reservoir is therefore conditioned by the value of the $c_{threshold}$.

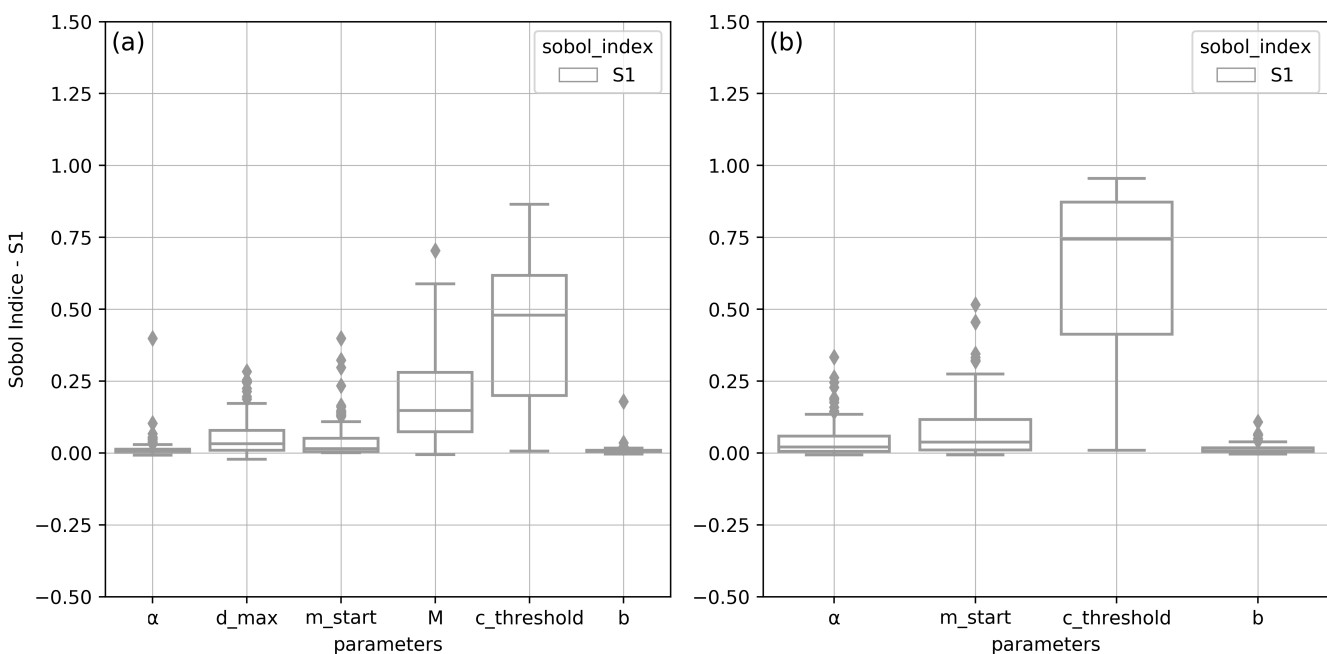

**Figure 6.** Distribution of first (S1) order Sobol indices on the modelled reservoirs for each parameter : (a) in irrigation and (b) in non-irrigation reservoirs.

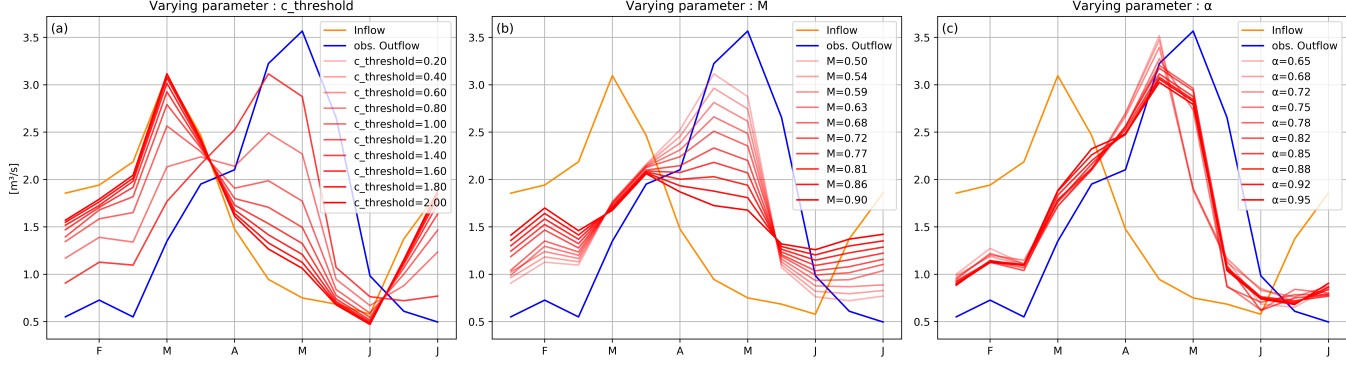

**Figure 7.** Example of seasonal pattern sensitivity of 'Gonzalez Lacasa' reservoir outflow to three of the DROP model parameters : (a) $c_{threshold}$, (b) $M$ and (c) $\alpha$. The remaining parameters were to their default values (see Table 2).

The influence of $M$ on the the minimum release is shown in Fig. 7(b). The higher the value of $M$, the greater is the part
of mean inflow set as minimum release and the lower is the remaining part of water dedicated to meet irrigation needs during
peak demand. Since the *'Gonzalez Lacasa'* reservoir is considered to have a relatively large storage capacity (with $c_{threshold} =$
0.5 as default value) among the considered reservoirs, the outflow will still follow the seasonal curve of water demand, but $M$





will influence release peak by extending or flattening the outflow curve to maintain a minimum release level required through the year. For run-of-river dams, $M$ is irrelevant since the release follows the monthly inflow. $d_{max}$ also controls the variation

of outflows over the year in the same way, but only when the DPI ratio is low. Its impact remains limited beyond the (1-$M$) threshold, as the release curve is fixed to maintain the required minimum outflow.

The parameter $\alpha$, on the other hand, does not affect the long-term mean pattern of release. In fact, $\alpha$ only operates on outflows on an inter-annual basis, through $K_y$, to offset the excess or shortage of stored water from one year to the other, especially when the critical filling zones are reached (reservoir in dead storage zone or overflowing) in order to bring the reservoir water

state to the ideal filling curve over the long term (Fig. 7 (c)). The same behavior is noted for $m_{start}$, which is not shown in Fig. 7.

The $c_{threshold}$, $\alpha$ and $m_{start}$ parameters have the same effect on non-irrigation reservoirs. The influence of each of the three parameters on [1979-2013] monthly mean outflows from the *'Alcantara II'* reservoir (GRanD ID 2800) in the Tagus river basin, which has a relative capacity of 0.6 and is mainly used for electricity generation, is shown in Fig. 8. For this specific

reservoir, the simulated period includes several wet years and there were periods when water flowed over the spillways during winter, which explains the alignment of the outflow with the inflow during this season.

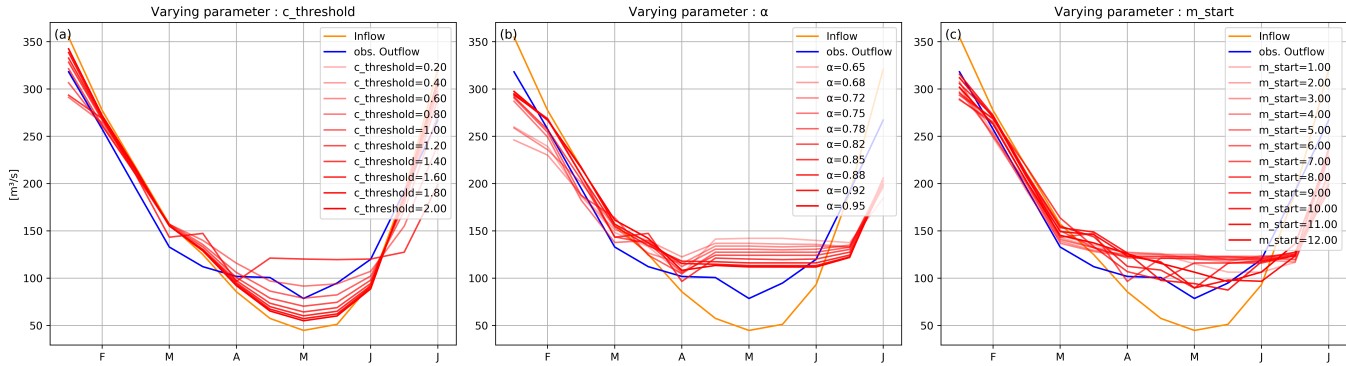

**Figure 8.** An example of the seasonal pattern sensitivity of the 'Alcantara II' hydroelectricity reservoir outflow to three of the DROP model parameters : (a) $c_{threshold}$, (b) $\alpha$ and (c) $m_{start}$. The remaining parameters were set at default values (see Table 2).

The distribution of total sensitivity indexes $ST$ (in grey), alongside with $S1$ (in white), of each parameter are shown in Fig. 9. A significant gap is observed between the first and total order index distributions. This confirms the non-negligible effect of the parameters' interactions on the output variance, involving mainly $c_{threshold}$. In the median, based on the $ST$ definition, $\sim$62%

of $C_{2M}$ variance in irrigation reservoirs, and $\sim$87% in non-irrigation reservoirs, are attributed exclusively to $c_{threshold}$ and its interactions with other parameters: it is indeed the most important parameter of the DROP model. $\alpha$ and $b$, on the other hand, were in the median low values, which makes them the least important parameters.

The distribution of second-order Sobol indices for each parameter couple for the irrigation and non-irrigation reservoirs considered separately, are shown in Fig. 10. The results reveal that the parameter with the most interactions overall with other

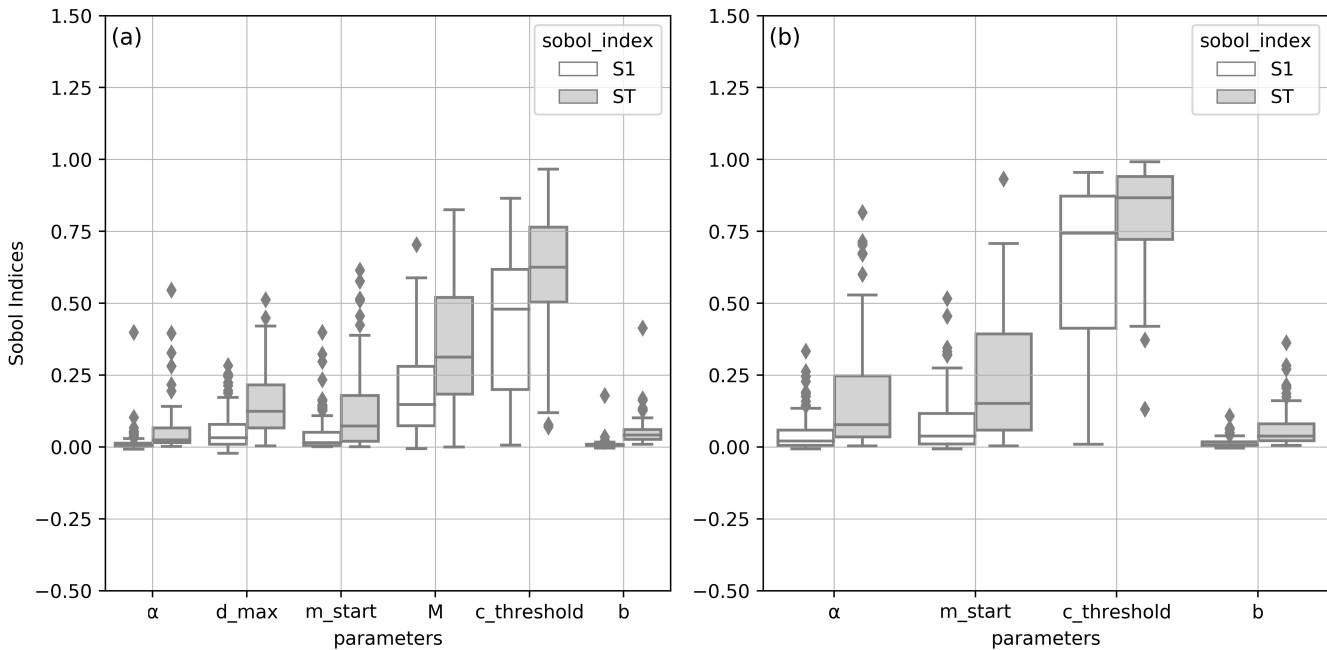

**Figure 9.** Distribution of first (S1) and total (ST) order Sobol indices on the modelled reservoirs for each parameter : (a) in irrigation and (b) in non-irrigation reservoirs

parameters is the $c_{threshold}$. This is mainly due to its position at the end of the model chain where provisional releases are corrected. For the rest of the parameters, the interaction between $m_{start}$ and $\alpha$ is more marked in non-irrigation reservoirs since the scheme is simplistic and the outflow depends mainly on the $K_y$ annual coefficient that is driven by both $m_{start}$ and $\alpha$. In irrigation reservoirs, the coefficients $d_{max}$ and $M$ interact because they control irrigation water demand and the reservoir water storage allocated to it. As they are both involved in the reservoir water balance, they jointly control the outflow. The $b$

coefficient meanwhile interacts only with $c_{threshold}$ when computing the demand-controlled release ratio R calculation. The total influence of $b$, mostly resulting from the interaction with $c_{threshold}$, is only seen in reservoirs with low relative capacities.

Since $c$ is a defining characteristic of each reservoir and is involved in the outflow computation, the first and total order Sobol indices were rearranged according to their reservoir corresponding $c$ values, here grouped into 6 ranges to simplify the presentation of the results to better identify the impact on parameter ordering and interactions (Fig. 11).

For low values of $c$ (less than 0.01), the $c_{threshold}$ is the only relevant parameter. The total influence of $b$, on the other had, is mostly related to its interaction with $c_{threshold}$. The remaining parameters are negligible since their first order Sobol indices are almost zero and the total order is very low. For reservoirs with a medium storage capacity ($0.01 \leqslant c \leqslant 1$), a significant part of total variance is due to parameter interactions, apart from $b$, as all Sobol indexes increase significantly from first to total order. This is even more noticeable when $0.1 \leq c \leq 0.3$. Above a relative capacity of 0.3, the $M$ parameter in irrigation

reservoirs gains in importance and increases interactions with the $c_{threshold}$ as $c$ is bigger. For non-irrigation reservoirs of the





same category, $c_{threshold}$ takes on more importance as the storage capacity increases (less spreading and very high $S1$). The remaining parameters lose relative importance. For the very high relative capacity values, $\alpha$ and $m_{start}$ are almost irrelevant, regardless of the operating objective. Finally, for nearly all reservoirs combined (except those with very small values of $c$), the $b$ coefficient has almost no influence and can therefore be set to a nominal value.

When using Sobol indices, the representation of model output uncertainty is limited to the variance only. Using the distribution function instead provides a complete description of uncertainty in the model output. The deviation of the $C_{2M}$ distribution function caused by two different parameters $c_{threshold}$ and $\alpha$ over the irrigation reservoirs is shown in Fig. 12. Here, the same samples generated for Sobol's index calculation were used. First, the unconditional PDF of the model output $C_{2M}$ (shown in black) is obtained when all input parameters are randomly sampled from their distributions. Then, for each input parameter,
conditional PDFs of $C_{2M}$ are computed following different value ranges from the total parameter distribution: here, five ranges are considered. The importance of each parameter is proportional to the magnitude of the conditional PDF deviation from the unconditional one.

The large dispersion of the unconditional PDFs (colored) shows the strong impact of $c_{threshold}$ in the model outlet in Fig. 12 (a). All PDFs are nearly aligned for $\alpha$, and thus the value of this parameter has very little significance on $C_{2M}$
distribution. PDF-based sensitivity indicators like "Kolmogorov", "Kuiper", "Delta" were used to measure the deviation from the unconditional PDF, and the results were consistent with the conclusions made with the Sobol index. DROP tends to have very poor performance scores for high values of $c_{threshold}$ (above 2.76) where all the reservoirs are considered as " run-of-river " and the release would be close to the lines up with inflow as shown in Fig. 12. This figure indirectly demonstrates the reservoir model contribution in terms of improving river flow representation in anthropized basins where the magnitude and seasonal
flow dynamics are significantly altered by large storage capacity reservoirs, especially those for irrigation purposes.

## 5  Discussion

### 5.1  Limits of the DROP model scheme

The DROP model outputs are affected by several uncertainties linked to the model inputs and the algorithm. The parameterized model is based on a generic scheme of reservoir operation, which inevitably implies a simplification in terms of water release.
Concerning the representation of operational purposes, the algorithm fails to differentiate between other purposes than irrigation and considers that a constant release is expected by the rest of the reservoirs. In addition, only the main objective of the reservoir is represented, and the releases of the multi-purpose reservoirs are not entirely represented since their management rules are more complex to describe. This explains the poor performance of the model at the level of reservoirs that are used for irrigation though it is not their main objective : DROP is very simplistic for the rest of its possible purposes so that the
seasonal shift in water discharge is not always well reproduced. This is the case of the reservoir "Los Bermejales" for example, a multi-purpose reservoir located in the basin of the Guadalquivir river, south of Spain, which is mainly used for water supply but it is also used to provide irrigation water demands(Fig. 13).





In addition, the model computes releases independently on each reservoir. The cascade of reservoir operations, which can be coordinated with each other, are thus not captured. More specific studies on multi-objective reservoirs Wu and Chen (2012); Wang et al. (2019) and multi-reservoir systems have been conducted Chang et al. (2014); Tan et al. (2017); Rougé et al. (2021), but they all remain complex and are reservoir-specific. These methods were evaluated only at the local scale and are very difficult to extend to a global scale because they need a significant amount of observed input data and require detailed operating rule knowledge.

Regarding the representation of releases from non-irrigation reservoirs more generally, the scheme remains very simplistic since release policy is not driven by physical processes. In fact, operation rules of these types of reservoirs involve complex socio-economic and political factors that are different in each country. Simulating other management purposes are mainly based on optimization algorithms, as it is the case for hydro-power dam releases for instance, where the objective functions are economically-oriented (i.e. to maximize energy production) Moeini et al. (2011); Feng et al. (2017); Chong et al. (2021). These methods remain very reservoir-specific and are currently deemed to be too complex to be applicable at a large scale.

The model provides a relatively good performance in representing irrigation reservoir operations because of its physical approach that links water releases to crop water demands. However, some simplifications are to be noted which could be improved in the future. The irrigation water demand estimation is based on the irrigation scheme in ISBA LSM, which has its own limitations (Druel et al., 2021). Also, in this version of the reservoir scheme, water demands for each irrigation reservoir is reduced to considering pixels that are downstream of the reservoir at a given maximum distance $d_{max}$. This creates inconsistencies because water demand is not linked to the reservoir water storage capacity, which leads for some reservoirs to much higher water demands compared to water supply. Moreover, here there are no proportionality rules between reservoirs for irrigation grids that are located in shared command areas because the model runs on each reservoir independently. We end up repeatedly counting shared pixels and this leads to over-estimating water demands. According to results from the sensitivity analysis, parameter $d_{max}$ alone (ranged from 1 to 250 km) does not have much influence on outflow variance, and this is even more true when irrigation demands are considered excessive compared to inflow (DPI>=1-$M$) because in that case the amount of irrigation demand is not more significant in outflow computing, only the seasonal variation defines the reservoir release curve. Zhou et al. (2021) suggested an efficient way to overcome this issue by defining a least-cost adduction network, based on Portoghese et al. (2013) and Neverre et al. (2016), to connect demand pixels to abstraction points, either from reservoirs or rivers, by using topographic information, distance and upstream areas of the river abstraction points. The method is implemented in the routing model of ORCHIDEE (ORganizing Carbon and Hydrology In Dynamic EcosystEms; Nguyen-Quang et al., 2018) and can be easily implemented in other routing models. It is also interesting at this stage to account for all possible sources of water withdrawals, including underground water, canals, but also the abstractions made directly from the reservoirs' storage, in order to have a more realistic representation of reservoir releases.

Another aspect which is not explicitly simulated in the model is water abstraction. In this study, abstractions are taken into account indirectly since the inflow is reconstructed from observations at the inlet of each reservoir. But once the DROP model is implemented in a hydrological model, the tributaries inflows will correspond to river flows simulated by the routing model, therefore there should be a deterioration of the performance index on discharge with an error spreading along the anthropized





rivers. However, by coupling the above with a model that takes irrigation into account, such as the new version of the ISBA land surface model for example (Druel et al., 2021), the water releases from the reservoir model can be linked to crop irrigation
needs and thus river water withdrawals can be represented as well as those taken directly from the reservoirs.

## 5.2 Contributions of the sensitivity analysis to a clearer understanding of the DROP model

The sensitivity analysis has revealed the most influential parameters and those that can be set using pre-defined values without impacting the model output uncertainty distribution. It emerged that $c_{threshold}$ is the most influential parameter in representing reservoir releases, and this result is consistent with the analysis of Shin et al. (2019) on the role of the release ratio $R$ where
$c_{threshold}$ is involved. Indeed, the aforementioned study showed the positive contribution of this formulation on the $R$ ratio ($R = \alpha c$, i.e., $c_{threshold} = 1/\alpha$) and its optimization in improving the representation of release and stabilizing water storage in the simulated reservoirs, especially those with $0.21 \leq c \leq 1.18$ (the values of $c$ for which $4c^2 = \alpha c$ and $c = 1/\alpha$, $\alpha$ being set at 0.85).

Hanasaki et al. (2006) conducted a sensitivity test on the $\alpha$ parameter for a case study of a large relative capacity hydroelectric
reservoir ($c = 2.28$). The model was tested with 4 different values of $\alpha$ (0.65, 0.75, 0.85, and 0.95) and found that this parameter had low impact on the simulated release but high impact on the simulated storage, except when the reservoir is full. $\alpha$ showed high sensitivity to releases when events with water passing over spillways were more frequent. Their conclusions concerning the sensitivity to outflows are also in line with the results found here.

Actually, the sensitivity analysis undertaken in this study focused on the model sensitivity on the average representation of
releases rather than in the filling levels of reservoirs since the focus of the study is on flow representation and the effect of anthropogenic factors in altering the flow dynamics along the rivers. If we were to focus on water resource availability and water management issues, the sensitivity analysis should also focus on the uncertainty in representing water storage levels in the reservoirs, considering the $C_{2M}$ over volumes as the variable of interest. In this configuration, as shown on Fig. 14, the parameters $\alpha$ and $m_{start}$, alongside $c_{threshold}$, will emerge as the most influential parameters with significantly high sensitivity
indices of both first and total orders.

Both $\alpha$ and $m_{start}$ directly affect the bias in the filling curve representation. In the ideal case where the reservoir state is not on the boundary conditions, the filling curve is vertically shifted following the value of $\alpha$ and the chosen month, in order to bring the reservoir volume at the beginning of each operating year in line with the ideal filling rate $\alpha C$, as shown in Fig. 15, without changing the seasonal release pattern. The larger the seasonality of reservoir water levels, the greater is the effect of
both parameters, which is very noticeable in large-storage capacity reservoirs used for water supply, and hence explains the two parameters' interaction with $c_{threshold}$.

## 6  Conclusions

In this paper, a global parameterized model, DROP, was reconstructed based on the Hanasaki et al. (2006) generic scheme to represent reservoir releases in Spain. Results reveal the positive contribution of the model in representing the seasonal



cycle of discharge and storage variation, specifically for irrigation large-storage capacity reservoirs as the model succeeds in reproducing the seasonal shift between inflows and outflows, improving river flow representation ($C_{2M}$ improvement rate) by 123% in the median. Spain represented an idealized case study in terms of the data availability, but the context (in terms of water needs) is applicable to certain other regions. While in-situ observations are currently available to characterize Spain's reservoir dynamics, different remote sensing data will allow the extension of the model to any other river basin over the globe,

more specifically to those which are ungauged, since the reservoir model relies only on GRanD reservoir database (which is global). This work is in preparation for the upcoming SWOT wide-swath altimetry mission (Biancamaria et al., 2016), which will provide the data necessary to make improved global scale river and reservoir storage and flow estimates. Furthermore, the results highlight the importance of incorporating reservoir operations into large-scale hydrological models for a more realistic representation of river flows and thus the water mass exchange with oceans and the atmosphere. The physical approach of

DROP is consistent with that of ISBA-CTRIP LSM-RRM (Land Surface - River Routing model): irrigation demands used as input to the reservoir model can be simulated by the irrigation module recently integrated into ISBA (Druel et al., 2021) and CTRIP already includes a lake model "MLake" (Guinaldo et al., 2021) that a priori models inland water bodies at a global scale, calculates mass balance and lake outflow at the global scale, and provides the foundation for integrating man-made reservoirs operations. The next step is to implement the DROP model in MLake and create a link between the two anthropization models

by coupling the new versions of ISBA (irrigation) and CTRIP (reservoirs).

The sensitivity analysis, based on Sobol's method, was conducted on the $C_{2M}$ representation of the mean seasonal outflow patterns. The results show that the most important parameter overall is $c_{threshold}$. $M$ and $d_{max}$ are ranked second, in the median, in terms of irrigation reservoir release representation and their importance is linked to the reservoir relative capacity. $\alpha$ and $m_{start}$ have less influence on both types of reservoir outflow seasonal dynamics but are important if the focus is on

reservoir water storage values. The results represent an essential step to further improve either river flow modeling or reservoir water storage, through calibration schemes and assimilation of new remote sensing products, by targeting the most influential reservoir model parameters. Overall, integrating this reservoir model into LSM-RRMs, which are in turn coupled to climate and earth-system models (such as the CNRM-CM and CNRM-ESM; Voldoire et al., 2019; Séférian et al., 2019), will provide a major advance in understanding past climate reanalysis and will enable a more realistic representation of future scenarios

under climate change.

*Code and data availability.* The DROP model and the sensitivity analysis codes are available on Zenodo. Post-processing codes are also available. All information can be found in the following repository : (https://doi.org/10.5281/zenodo.6389405; Sadki, 2022).

*Author contributions.* MS, SM, AB and SR designed the study and determined the methodology. MS and SM developed the reservoir model and the sensitivity analysis algorithm. MS performed the analysis and wrote the original draft. All authors contributed to the editing and

review of the paper.





*Competing interests.* The authors declare that they have no conflict of interest.

*Acknowledgements.* We would like to thank Dr. Pere Quintana-Segui from the Ebro Observatory for providing the SAFRAN-based meteorological dataset for Spain which was used to force the ISBA LSM and compute irrigation demands. We also thank him for providing pre-processed products of the *in situ* natural and anthropized flow and volume observations over Spain.

This study is part of Malak SADKI's thesis work, co-funded by the French National Center for Space Studies (CNES) and the Occitanie Region in France.





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



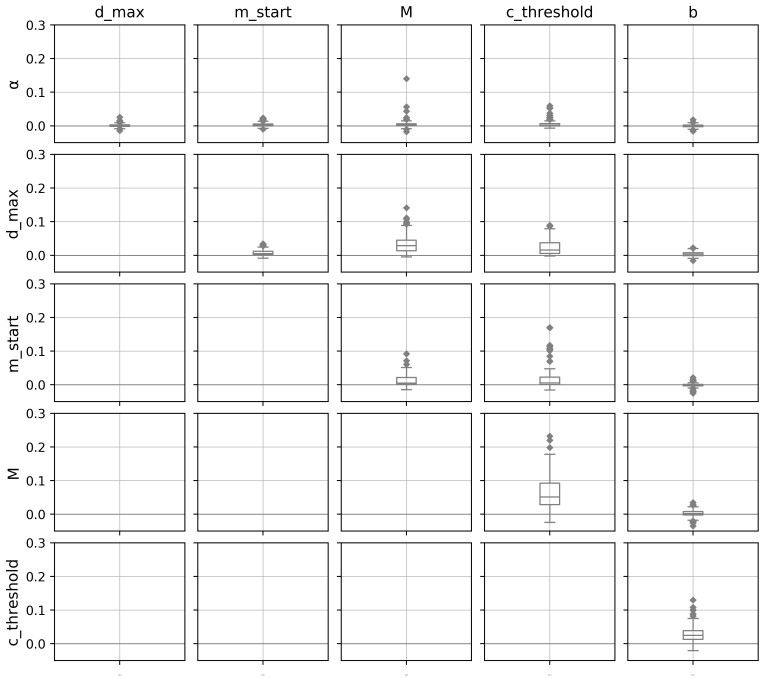

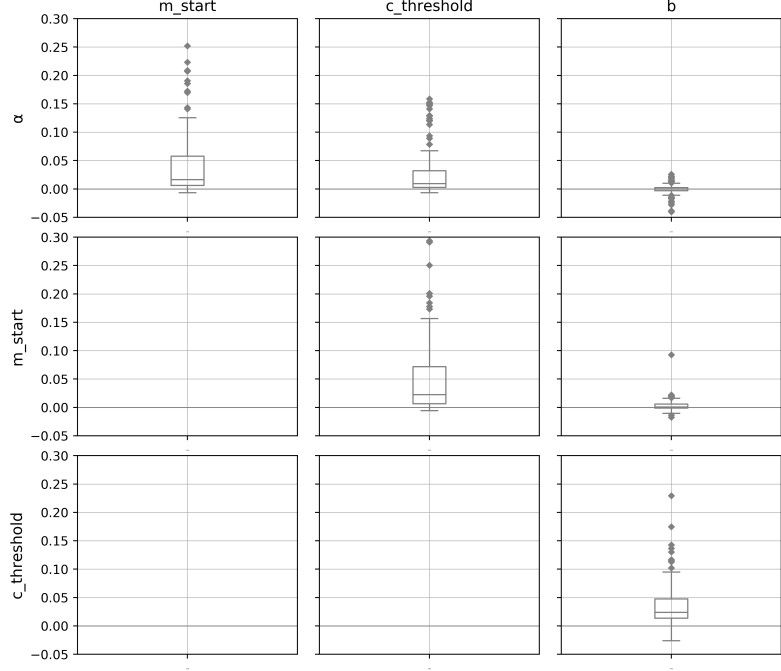

**Figure 10.** Distribution of second order Sobol indices (S2) on the modelled reservoirs for each pair of parameters : (a) in irrigation and (b) in non-irrigation reservoirs





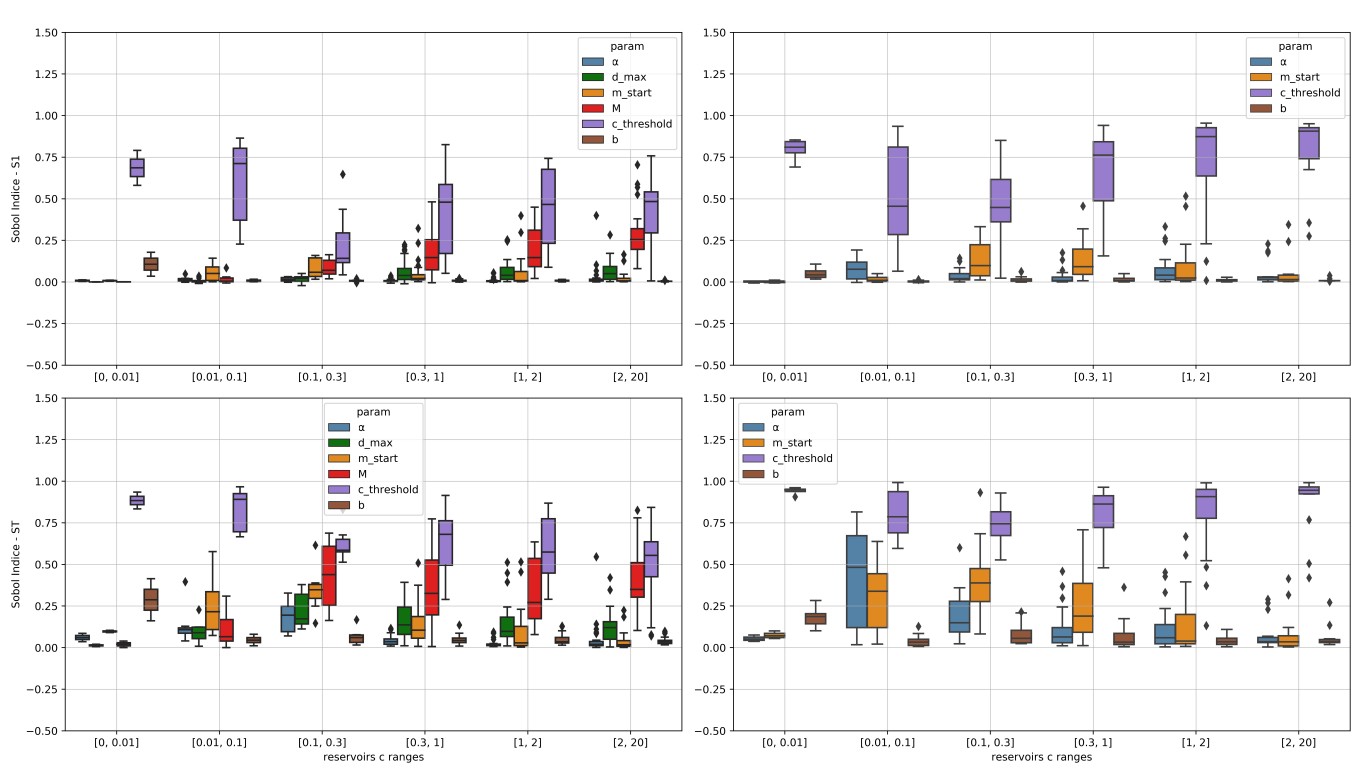

**Figure 11.** Distribution of first ('S1') and total order ('ST') Sobol indices of parameters according to relative capacity : (a),(c) in irrigation and (b),(d) in non-irrigation reservoirs.





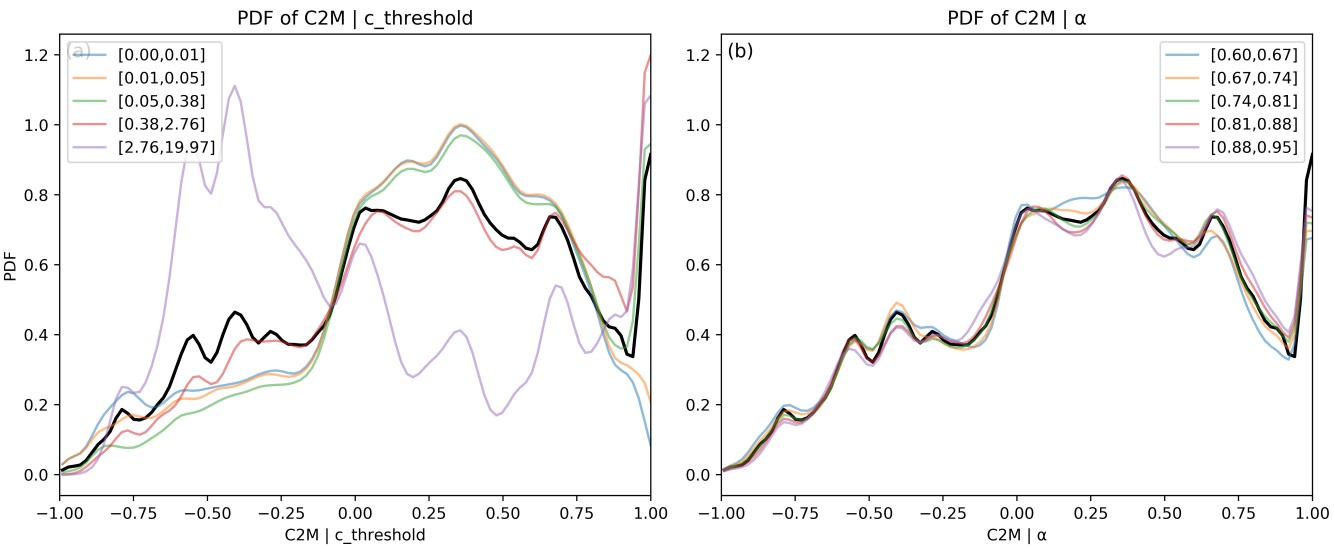

**Figure 12.** Shifts in $C_{2M}$' Probability Distribution Function (PDF) depending on $c_{threshold}$ (a) and $\alpha$ (b) range values, within irrigation reservoirs. The unconditional PDFs of the DROP model output $C_{2M}$, obtained when all input parameters are randomly sampled, are displayed in black. The conditional PDFs are shown in color depending on the parameter value range.

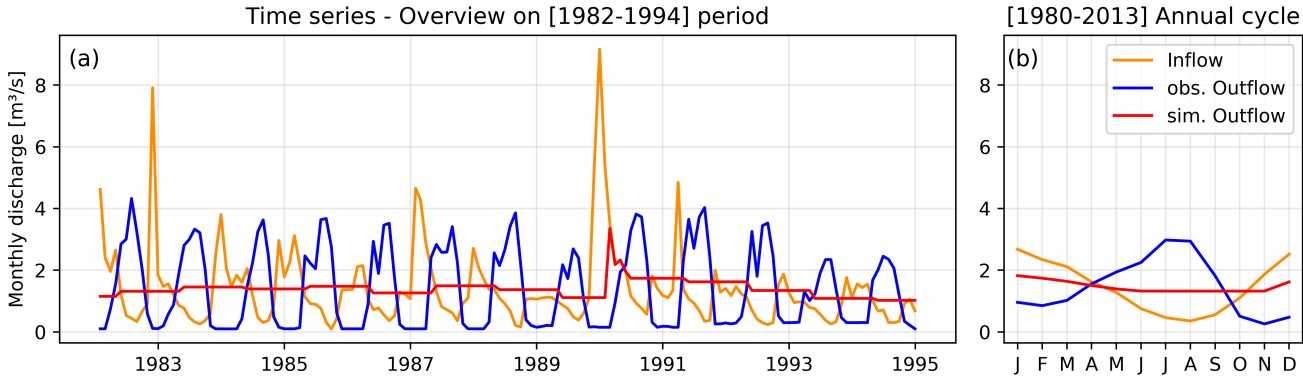

**Figure 13.** Example of time series (a) and monthly means (b) of simulated releases from the 'Los Bermejales' reservoir, a multi-purpose reservoir primarily used for water supply and which also operates to provide irrigation water demands. The reservoir model failed to reproduce observed irrigation releases as it does not consider secondary objectives.





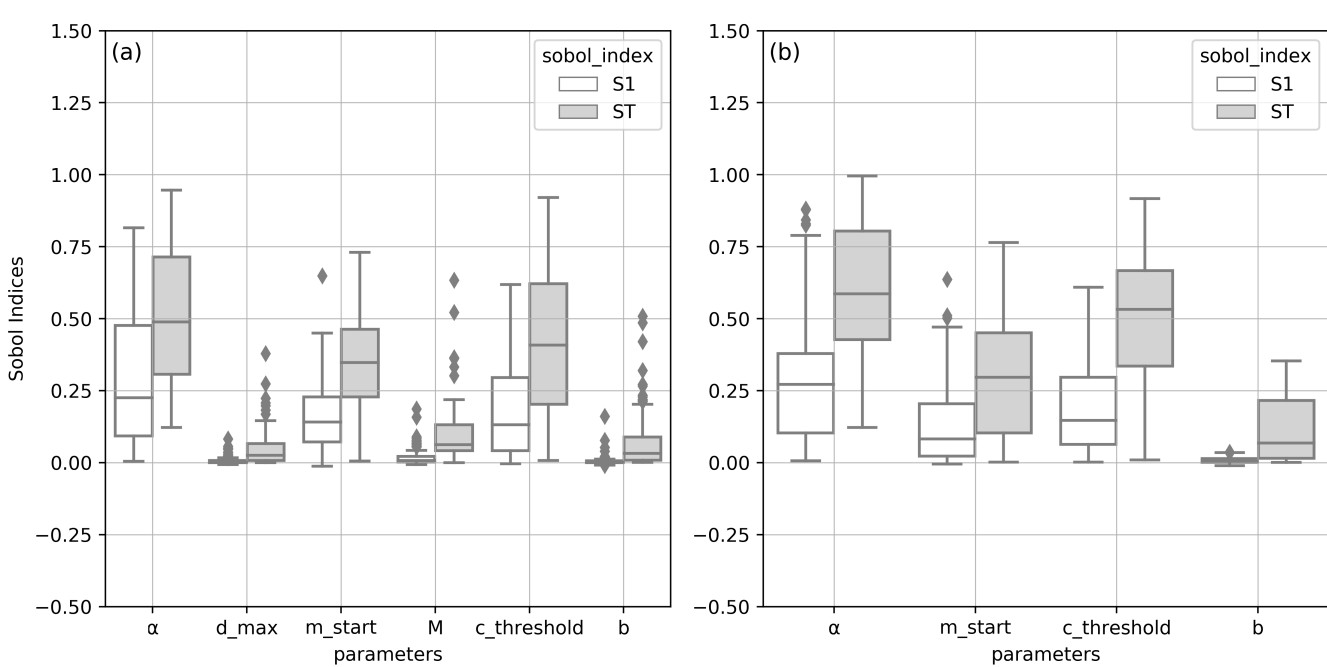

**Figure 14.** Distribution of first ('S1') and total order ('ST') Sobol indices of parameters based on $C_{2M}$ variance over volumes : (a),(c) in irrigation and (b),(d) in non-irrigation reservoirs.





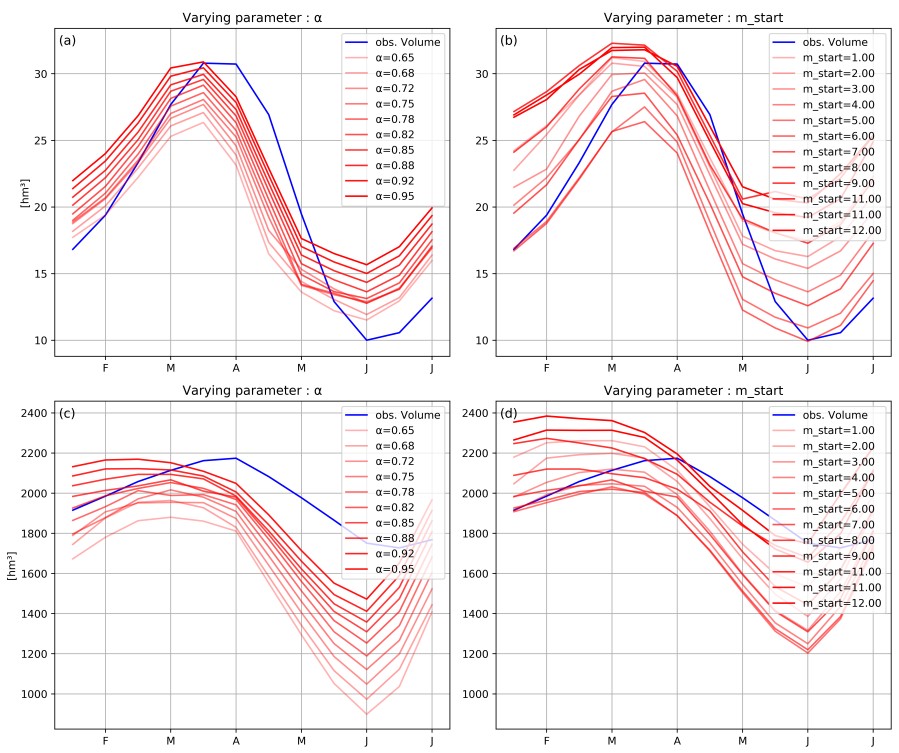

**Figure 15.** Example of seasonal pattern sensitivity of "Gonzalez Lacasa" (a,b) and "Alcantara II" (c,d) reservoir volumes to two of the DROP model parameters : (a,c) $\alpha$ and (b,d) $m_{start}$. The remaining parameters were set at default values (see Table 2).