# Peer review of "Implementation and sensitivity analysis of a Dam-Reservoir OPeration model (DROP v1.0) over Spain"

_Geoscientific Model Development, 2022_

## Author Comment (AC1)

**Response to the comments about the submitted paper Implementation and sensitivity analysis of a Dam-Reservoir OPeration model (DROP v1.0) over Spain**

We would first like to thank the reviewers for the time spent carefully reading our submitted manuscript and for writing these very constructive reviews. Considering the different comments and recommendations addressed to us, we have modified the paper accordingly and have provided a point-by-point response to each review.

Please note that reviewers' comments are in italics while our answers are not. Additions to the original manuscript are indicated in blue. The line numbering mentioned in this document refers to that of the revised paper.

We hope that we have provided the desired explanations for the sections concerned and the needed supplemental material. We hope that we have improved the manuscript to meet GMD standards after this revision.

We look forward to hearing from you in due time regarding our submission and to respond to any further questions and comments you may have.

Sincerely,

Malak Sadki malak.sadki@meteo.fr CNRM, Université de Toulouse, Météo-France, CNRS, Toulouse, France

**Answers to Reviewer 1**

**Comment R1.1** [General comment] : This manuscript is overall well written and easy to follow. The topic fits GMD very well. The authors nevertheless are challenged to better highlight its novelty or significance beyond applying an existing reservoir scheme over Spain. The objective of this study was two folds. First, it "proposes" a global and parameterized reservoir model. Secondly, it performs sensitivity analysis on the model parameters. Specific comments on the two aims are the following.

Answer to R1.1 We thank you very much for your comments and understand your reservations regarding some aspects of the paper. We have carefully considered the specific comments and have addressed them one by one. We hope to have provided a clearer explanation of the choices made and that we have demonstrated the added value of the work carried out.

**Comment R1.2** [Specific comment - The first aim of the study] : Their statement is factual as the model simply converts deterministic values presented in Hanasaki et al. (2006) to parameters. However, even that is not the first attempt as it has been attempted by Shin et al. (2019). For example, in Equation (5), at M = 0.5, this Equation becomes Hanasaki. Similar case for Equation (6). Therefore, in general, this is an existing scheme. The naming of a scheme developed by others as new is also wrong from my point of view.

Answer to R1.2 Indeed, the reservoir model described in this paper uses some of the parameterization done by [Shin et al., 2019], as clearly indicated in the paper. However, the version presented hereby proposes a full parametrization by pointing out all the parameters involved in the model run. For instance, the starting month of the operational year, noted  $m_{start}$  here, is clearly identified as a model parameter as it impacts the annual coefficient  $K_y$  and therefore the real monthly release  $r_m$  in equation (6). The same holds for  $d_{max}$  (which was introduced in the current version), since it affects the DPI, and thus the formula to be used to compute provisional release  $r'_m$  (equation (5)). A more explicit parametrization of R is also an advantage compared to what is proposed in [Shin et al., 2019], because it generalizes the old formulations and brings out the parameter  $c_{threshold}$ . The latter was set to 0.5 and  $1/\alpha$  by [Hanasaki et al., 2006] and [Shin et al., 2019] respectively. This parametrization of R enables to distinguish the dual role of  $c_{threshold}$  in equation (6) from that of coefficient b, which only describes the way the transition is made between a demand-controlled operating rule and an inflow-controlled one.

Concerning the naming of the reservoir model, we agree on this point. In fact, the first submitted version of the paper did not include a name for the model, since it is considered to be a generalization of the Hanasaki model (as referred to in section 2.1) through a more comprehensive parameterization. The naming was done following the request of the topical editor to respect the GMD policy so that the paper could be put into preprint. The reservoir model presented herein is to be implemented into our global river-lake-reservoir-groundwater hydrological scheme CTRIP, thus it represents the foundation for further future developments which we are already undertaking, in particular regarding the non-irrigation reservoir equations (description of a parameterized filling curve instead of a constant line currently used). So the a posteriori model will diverge from the [Hanasaki et al., 2006] model. More references of [Hanasaki et al., 2006] and [Shin et al., 2019] have been added in the article.

**Comment R1.3** [Specific comment - The second aim of the study] : Concerning the sensitivity, the identified most sensitive parameters are ones anyone could generalize through educated guesses.  $c_{threshold}$  is the parameter for the lower bound of the Hanasaki schemes, the storage capacity to mean total annual inflow ratio. Equation (6) represents the final release where 'c' is the determinant factor. Below the threshold, the Equation is a one-parameter function, and the threshold has no influence. Above the threshold, it is a three-parameter and uses the threshold value directly. This is also the case for the M parameter (Equation 5). Further, the sensitivity analysis is performed on the flow; both M and  $c_{threshold}$  play a major role in release estimation. So, one can easily reach the same conclusion without performing the sensitivity analysis.

Answer to R1.3 We agree with the reviewer that educated guesses and first sensitivity tests can easily show that  $c_{threshold}$  and M are likely to be the most influential parameters. However, we believe that more relevant findings could be drawn with the contribution of the sensitivity analysis, provided in this article, beyond the previous sensitivity tests performed on some of the parameters separately.

Firstly, the sensitivity analysis allowed us, in addition to confirming this insight, to quantify the influence of each parameter uncertainty on either release discharge or volume. For each parameter, the influence of the parameter taken alone could be differentiated from the one due to its interaction with the others, as shown in Figure 10 (second-order Sobol indices (S2) being represented for each pair of parameters). The parameters are thus sorted in order of priority and by type of influence, and this result is more difficult to guess without running a comprehensive sensitivity analysis.

Secondly, it has also been shown that there is a difference in the influence of the parameters depending on whether one is interested in the representation of outflows, and therefore in flow dynamics and reservoir volume variations, or whether one is interested in absolute values of reservoir storage, which relates to water resource management questions.

The sensitivity analysis also shows that the influence of these parameters and their interactions evolve according to the relative capacity of the reservoirs, as shown in Figure 11. And this work provides a reference for further studies, as the significance of the model's parameters will vary according to the range of reservoir relative capacities being studied. The scheme is planned to be used within a data assimilation framework, and normally such an analysis is very useful for quantification of errors for such an application.

To better highlight the interest of the sensitivity analysis within the paper, we have added the lines (68-70) in the introduction and have revised the lines (542-547) in the conclusion as follows:

68-70 : "In this regard, the study aims to quantify the influence of each parameter and to reveal the types of influence that each of them holds by dissociating individual effects from

possible interactions."

542-547 : "It has also been proven that the significance of the model's parameters varies according to the range of reservoir relative capacities being studied. The results represent an essential step to further improve either river flow modeling or reservoir water storage, through calibration schemes and assimilation of new remote sensing products, by targeting the most influential reservoir model parameters. This work provides future studies with a fully generalised parameterization of the reservoir scheme along with a deeper insight into the way each of the parameters influence the model outputs and how their importance changes depending on the reservoir characteristics and the output variable of interest."

**Comment R1.4** [General comment] : Since the novelty on the modeling or parameterization side is somehow limited, perhaps the authors can provide more analysis and discussions on the similarity and difference between the reservoir/river dynamics in Spain and other regions, hence producing improved scientific understanding.

Answer to R1.4 We are fully aware that the reservoir model presented is largely based on an existing one, which is extended by a fully generalized parameterization. Nevertheless, we believe that the study has an interesting contribution when combining the proposed full parameterization with a sensitivity analysis, as it will provide future studies a deeper understanding of how each parameter affects the different model outputs and the way their importance may vary depending on the reservoir characteristics and the output variable of interest. It has been proven that the distribution of the relative capacities of the reservoirs being studied and the subject of the study (flow dynamics or quantification of water storage) are to be taken into account when choosing the parameters to be set.

Moreover, the sensitivity analysis is a further important step in order to implement regional calibrations and remote sensing products assimilation in large-scale hydrological models, since it enables to target only the most influential parameters and thus considerably reduces the computation time and costs. The paper presented an application of the reservoir model over Spain specifically since it has highly anthropized basins that have been relatively poorly studied at the national scale.

A calibration scheme was recently run over the reservoir model parameters within Spain reservoirs. Regional parameters for the case study were set upon this step and a comparison to default values from [Hanasaki et al., 2006], [Biemans et al., 2011] and [Shin et al., 2019]  $(R_{new} \text{ simulation}, \text{ where } (c_{threshold}, b) = (1/\alpha, 1))$  was performed. The findings can be included to complete the study and to show better the added value of the sensitivity analysis. We propose to include these calibration results in the paper as a supplementary material (see proposed supplementary material).

**Answers to Reviewer 2**

**Comment R2.1** [General comment] : The authors examined parameterizations of reservoir operations in Spain. They employed a series of formulations developed by Hanasaki et al. (2006) that reproduce reservoir release and storage. They conducted intensive parameter sensitivity tests and identified several highly sensitive parameters. They also analyzed the performance of 215 reservoirs to discuss why they performed better or worse. The formulation of Hanasaki et al. (2006) has been employed in many global hydrological models, but is rarely additionally and carefully validated. This study is unique because it presents the results of intensive validation in 215 reservoirs in Spain where reliable data are well accessible. Another interesting result, somewhat surprisingly, is that the simple formulation works well with observations in many cases. The manuscript is quite well written and clear. I only noticed that a few technical clarifications are necessary in several parts as detailed below.

Answer to R2.1 Thank you very much for your comments and for pointing out the added value of this study. We have read the specific comments carefully and addressed them one by one.

**Comment R2.2** Line 82 "direct runoff Rd": What is this? Is this different from "tributary inflows"?

Answer to R2.2 We define direct runoff,  $R_d$ , as water that runs off the ground surface after rainfall or snowmelt and which, having not infiltrated through the soil, runs off the surface to the reservoir storage. The tributary inflows,  $Q_{trib}$ , relate to the streamflows of the main river or the various tributaries that reach the dam storage reservoir.

We have revised the lines (82-85) as follows :

82-85 : " $Q_{in}$  and  $Q_{out}$  stand respectively for the net inflow to the reservoir and its outflow at the outlet. In fact, the net inflow,  $Q_{in}$ , combines different physical processes, as shown in Equation (2): it includes water inputs from precipitation P falling directly into the water surface, direct runoff  $R_d$  representing water flows running off the surrounding ground surface that also feeds the reservoir, and tributary streamflows  $Q_{trib}$  flowing into the reservoir. Evaporation E and groundwater exchange  $Q_{qw}$  are also included in  $Q_{in}$ ."

**Comment R2.3** Line 135 "Unlike Hanasaki et al. (2006) scheme...": It is a bit unclear as to what exactly is different from Hanasaki et al. (2006). Does it mean that industrial and domestic water requirements are not considered in this study? For readers' convenience, it would be nice if the authors add a list of the major changes from Hanasaki et al. (2006).

Answer to R2.3 Water demands used in our version of the model only account for irrigation demands.  $d_m$  and  $d_{mean,y}$  in equation (5) are in fact monthly and annual mean irrigation water demands since this equation is specific to irrigation reservoirs. Industrial and domestic

water demands are not taken into account as they are highly uncertain and hardly accessible, as explained in lines (137-145) of the manuscript. This is different from what has been done in [Hanasaki et al., 2006] where time-constant country-fixed average values are included in water demands when computing the provisional release of irrigation reservoirs.

Following your recommendation, we suggest adding a list of the main changes from [Hanasaki et al., 2006] and [Shin et al., 2019] at line (164) of the manuscript as follows :

"The modifications brought to the model from the previous version of Hanasaki et al. [2006] and the Shin et al. [2019] parameterization can be summarized in the following list:

- The starting month of the operational year, which was calculated in previous versions at the level of each reservoir based on observed inflows and dam releases, is considered in this model version as a parameter, noted  $m_{start}$ . This overcomes the challenge faced by the old versions when applying the model in ungauged basins. Within our version,  $m_{start}$  is to be set separately for irrigation and non-irrigation reservoirs.
- The same applies to  $d_{max}$ , initially set to a constant value depending on the spatial resolution of the river routing model in which the scheme is being implemented in each study. Here,  $d_{max}$  is defined as a parameter to be set for reservoir command area delimitation.
- $d_m$  and  $d_{mean,y}$  in equation (5) are monthly and annual mean irrigation water demands, respectively. Industrial and domestic water demands are not taken into account, unlike in Hanasaki et al. [2006]
- *M*, set as 0.1 in Biemans et al. [2011] and Shin et al. [2019] schemes, and as 0.5 in Hanasaki et al. [2006] paper, is considered as a parameter in this version, keeping the same notation used in Shin et al. [2019] study.
- A more explicit parameterization of the demand-controlled release ratio R is provided hereby compared to what is proposed in Shin et al. [2019] paper. The generalized formulation of R in Equation 7 highlights the last two parameters of the model present version. This enables to distinguish the dual role of  $c_{threshold}$  in Equation 6 from that of the coefficient b, only describing the transition made between a demand-controlled and an input-controlled reservoir. Hanasaki et al. [2006] and Shin et al. [2019] have set  $(c_{threshold}, b)$  to (0.5,2) and  $(1/\alpha, 1)$  respectively.

A description of the reservoir model parameters is summarized in Table 1."

**Comment R2.4** Line 400-415 "When using Sobol indices, ...": I couldn't follow the discussion for these two paragraphs. In short, I couldn't understand what Figure 12 is showing. What does it mean by "shifts in  $C_{2M}$ "? What is the PDF being discussed here? A bit more detailed and readable explanation is needed.

Answer to R2.4 In this section, we added a moment-independent sensitivity method to validate the results found with Sobol indices. In fact, by looking at the  $C_{2M}$  distribution, the variance isn't fully representative of the full statistical characteristics (moments) of the distribution and therefore variance-based indices do not describe the complete output variability [Song and Wang, 2021].

Here, a further step is added to evaluate the sensitivity of parameters by assessing the influence of the entire parameter distribution on  $C_{2M}$  entire output distribution without reference to a specific moment of the output [Borgonovo, 2007, Chun et al., 2000, moment independence]. This method relies on different sensitivity indices, called moment-independent measures like Delta, Kolmogorov, Kuiper. Those indices compare the unconditional PDF to the conditional ones, each created by setting the studied parameter to a definite value range from its total distribution (Table 2). The more differences (also called shifts or deviations) between the PDFs, the greater the influence of the parameter. Since these indices do not separate main, total effect and interaction, this PDF-based method has not been used as the main approach of sensitivity analysis in this paper and is instead only used to validate the overall conclusions from Sobol indices.

To provide a better introduction and explanation of this part of the results, we have revised the lines (420-425) with the following :

420-425 : "When using Sobol indices, the representation of model output uncertainty is limited to the variance only which isn't fully representative of all the statistical characteristics (or moments) of the  $C_{2M}$  distribution. Using the distribution function instead provides a complete description of uncertainty in the model output. Here, we evaluate the sensitivity of parameters by assessing their influence on  $C_{2M}$  entire distribution without reference to a specific moment of the output [Borgonovo, 2007, Chun et al., 2000, moment-independent methods]. This method is used as a validation step for the overall conclusions found with the variance-based Sobol method."

**Comment R2.5** Line 452 "Zhou et al. (2021) suggested an efficient way to overcome this issue...". I couldn't fully understand what issue is focused on here and how Zhou et al. (2021) solved it. It would be helpful for readers if you could elaborate further. Perhaps, I guess this part discusses the cells that should be included in the estimation of irrigation water demand. The spatial resolution of Hanasaki et al. (2006) was 1 degree by 1 degree, or 110 km by 110km at the equator. Therefore, it seemed that summing the grid cells direct downstream would be sufficient to estimate irrigation water demand for large reservoirs. However, the spatial resolution in this study is 5km by 5km. It seems necessary to include not only the direct downstream grid cells, but also the surrounding ones. Some of the latest global hydrological models with a high spatial resolution have started to transfer water from main stream to the surrounding grid cells (e.g. the "aqueducts" of Hanasaki et al. 2018, 2022). Maybe such a concept is needed to estimate downstream water demand reasonably.

Answer to R2.5 The method used to delimit the control area of reservoirs in order to calculate water demands allocated to each of them, is based on the simple accounting of downstream irrigation grid cells at a given maximum distance,  $d_{max}$ , from the reservoir.

Since no proportionality rule is set between reservoirs with common irrigation grid cells, they are recorded on multiple reservoirs when their command areas are overlapping. This leads to an overestimation of the water demand at the level of each reservoir.

One solution suggested by Zhou et al. [2021] is to define a network that connects each irrigation pixel to a unique abstraction point, either a river or a reservoir, such that it is made sure the reservoirs command zones do not overlap anymore and that each pixel irrigation demand is counted only once. This approach doesn't only look at downstream grid cells but at all of the surrounding ones as the connections are drawn between reservoirs and irrigation pixels by considering altitude difference and distance in such a way that the adduction network has the lowest cost (optimized least-cost function).

Integrating implicit aqueducts into the river network, as carried on by Hanasaki et al. [2022], would improve the overall water transfers including in dams water storage and this would lead to a better estimation of supply and demand, but this aspect can be explored at the level of CTRIP river network once the reservoir model is included.

We have revised lines (470-473) and lines (476-481) to incorporate more details on the model's limitations in irrigation demand estimation, as follows:

470-473 : "Moreover, here there is no proportionality rule set between reservoirs with common irrigation grid cells. They are recorded on multiple reservoirs when their command areas are overlapping because the model runs on each reservoir independently. We end up repeatedly counting shared pixels and this leads to over-estimating water demands at the level of each reservoir."

476-481 : "Zhou et al. [2021] suggested an efficient way to overcome this issue by defining a least-cost adduction network, based on Portoghese et al. [2013] and Neverre et al. [2016], to connect each irrigation grid cell to a unique abstraction point, either a river or a reservoir, by using topographic information, distance and upstream areas of the river abstraction points. This approach has the advantage of considering not only the downstream grid cells, but all the surrounding ones. Most importantly, it ensures that reservoir command areas do not overlap anymore and that each pixel irrigation demand is only counted once."

**Answers to Editors : Request from Polina Shvedko**

**Comment AC.1** Please adjust the list of corresponding authors in the MS Records system and in the \*.pdf manuscript for the next revision

Answer to AC.1 The list of corresponding authors in the MS Records system has been changed to be consistent with the one in the revised pdf manuscript.

**References**

- H. Biemans, I. Haddeland, P. Kabat, F. Ludwig, R. Hutjes, J. Heinke, W. von Bloh, and D. Gerten. Impact of reservoirs on river discharge and irrigation water supply during the 20th century. *Water Resources Research*, 47(3), 2011.
- E. Borgonovo. A new uncertainty importance measure. *Reliability Engineering & System Safety*, 92(6):771–784, 2007.
- M.-H. Chun, S.-J. Han, and N.-I. Tak. An uncertainty importance measure using a distance metric for the change in a cumulative distribution function. *Reliability Engineering & System Safety*, 70(3):313–321, 2000.
- N. Hanasaki, S. Kanae, and T. Oki. A reservoir operation scheme for global river routing models. *Journal of Hydrology*, 327(1-2):22–41, 2006.
- N. Hanasaki, H. Matsuda, M. Fujiwara, Y. Hirabayashi, S. Seto, S. Kanae, and T. Oki. Toward hyper-resolution global hydrological models including human activities: application to kyushu island, japan. *Hydrology and Earth System Sciences*, 26(8):1953–1975, 2022.
- N. Neverre, P. Dumas, and H. Nassopoulos. Large-scale water scarcity assessment under global changes: insights from a hydroeconomic framework. *Hydrology and Earth System Sciences Discussions*, pages 1–26, 2016.
- I. Portoghese, E. Bruno, P. Dumas, N. Guyennon, S. Hallegatte, J.-C. Hourcade, H. Nassopoulos, G. Pisacane, M. V. Struglia, and M. Vurro. Impacts of climate change on freshwater bodies: quantitative aspects. In *Regional Assessment of Climate Change in the Mediterranean*, pages 241–306. Springer, 2013.
- S. Shin, Y. Pokhrel, and G. Miguez-Macho. High-resolution modeling of reservoir release and storage dynamics at the continental scale. *Water Resources Research*, 55(1):787–810, 2019.
- S. Song and L. Wang. A novel global sensitivity measure based on probability weighted moments. *Symmetry*, 13(1):90, 2021.

X. Zhou, J. Polcher, and P. Dumas. Representing human water management in a land surface model using a supply/demand approach. *Water Resources Research*, 57(4): e2020WR028133, 2021.

---

## Author Comment (AC2)

*Supplementary material to*

**Implementation and sensitivity analysis of a Dam-Reservoir OPeration model (DROP v1.0) over Spain.**

Malak Sadki[1], Simon Munier[1], Aaron Boone[1], and Sophie Ricci[2]

[1]CNRM, Université de Toulouse, Météo-France, CNRS, Toulouse, France
[2]CECI, CERFACS/UMR5318 CNRS, Toulouse, France

**Correspondence:** M. Sadki (malak.sadki@meteo.fr)

**1 Supplementary study - Reservoir model calibration over Spain**

To better underline the importance of the sensitivity analysis, a calibration scheme was run over the reservoir model parameters for Spain reservoirs. Model parameter sampling and $C_{2M}$ performance index optimization scheme are here performed using
5  the differential evolution stochastic method, available in open-source Scipy Python library (SciPy, 2022). An initial population of parameters values is generated following Latin Hypercube sampling in order to scatter the sample points as uniformly as possible over the parameter space and maximize its coverage (here the same bounds as for the sensitivity analysis, see Table 2 in Sadki et al. (2022)). A population size of 50 is chosen to ensure convergence of the calibration scheme. The algorithm seeks to minimize a cost function defined in this study as the difference between 1 and $C_{2M}$. The tolerance threshold is set
10  to 0.01, as given by default by the function. Regional parameters (same parameter values for all reservoirs) for the case study were set within this step and a comparison to default values from Hanasaki et al. (2006), Biemans et al. (2011) and Shin et al. (2019) ($R_{new}$ simulation, where ($c_{threshold}$, $b$)=(1/$\alpha$,1)) is hereby represented. The optimisation was carried out on reservoir outflows, independently on irrigation and non-irrigation reservoirs as the parameters involved are different depending on the main purpose (see section 3.4 in Sadki et al. (2022)).
15  Fig. 1 shows the distribution of parameter values before and after calibration, for irrigation and non-irrigation reservoirs separately, since the number of parameters involved is not same in both categories of reservoirs (6 and 4 parameters respectively). First, the parameter values used in the reservoir model default configuration (see Table 2) are shown in blue. In orange are shown default values used by Hanasaki et al. (2006), Biemans et al. (2011) and Shin et al. (2019) ($R_{new}$ simulation) studies. The remaining values represent two distinct optimisations: the points in red represent the configuration of parameters to best
20  simulate the whole of Spain's reservoirs. These represent the regional model configuration to be set for Spain. If we optimized each reservoir separately, the boxplots in black illustrate the spread and the range of the parameters' values. Regarding the results of the calibration over the entire country: the default global values of $\alpha$ and $M$ given by the previous papers are not appropriate for Spain; indeed, for $\alpha$, the optimal regional value ranges between 0.71 and 0.77 when considering all reservoirs. The calibration per reservoir even shows that half of the reservoirs simulate best flow seasonal variation with an $\alpha$ below
25  0.69. This may be related to the high anthropogenic pressure on water resources combined with the semi-arid climate, which

[Figure]

**Figure 1.** Parameter values before and after $C_{2M}$ optimization: In blue is represented as 'default' the set of parameter values used in the default parameterization of the given reservoir scheme. In orange are shown the default values used in (Hanasaki et al., 2006), (Biemans et al., 2011) and (Shin et al., 2019) ($R_{new}$ simulation) papers, labeled respectively as 'H06', 'B11' and 'S19'. The red marked 'opt. all' values represent the optimal parameters values when calibrating over all Spain reservoirs outflows. The boxplots in black represent the spread of optimal parameter values for each reservoir when optimized individually.

would make reaching 85% of storage an unrealistic target for this particular region. Concerning the parameter $M$, specific to irrigation reservoirs, the default values set by H06 and B11 are within the range of $M$ optimal values, but the optimal setting for the country's reservoirs is relatively higher (0.57). The optimal value of $c_{threshold}$ in both types of reservoirs is $\sim 0.56$ which is close to default values set in older versions. $c_{threshold}$ is the most influential parameter, the dispersion observed on

30    the optimums of this parameter when running a reservoir-by-reservoir calibration shows that setting a global or regional value limits the performance of the model. This is shown in Fig. 2 where distribution of the reservoir model performance index ($C_{2M}$) is displayed for both types of reservoirs.

[Figure]

**Figure 2.** Distribution of the reservoir model performance index $C_{2M}$ over outflows before and after calibration in irrigation (green) and non-irrigation reservoirs (light grey). The default configuration stands for the default setting of the model, the two others are respectively the model calibration performed uniformly on all reservoirs and on each reservoir individually.

The individual reservoir calibration, on the other hand, shows a wide dispersion of the optimal parameter values, especially for $\alpha$ and $c_{threshold}$ (Fig. 1). This is linked to the reservoir characteristics, but also to Pareto fronts that may occur, such as for

35    $c_{threshold}$, interacting with several other parameters, as the 2nd Sobol indices revealed in Figure (10). By taking into account the individual characteristics of each reservoir in the calibration, the performance of the model is considerably improved (Fig. 2). Overall, confronting Spain's optimal parameter ranges with the global default values shows the interest of having a regionalization which implicitly integrates the impact of the specific climate, land use and demographic pressure of a study area in the management of its dams and water resources. The dispersion of the optimums from the per reservoir calibration

40    and the significant performance gained show the importance of having specific values for each reservoir. To this purpose, observations from recent and future satellite missions, especially in ungauged basins. In particular the forthcoming SWOT which will provide water level observations and river flow estimates, as well as reservoir water surface areas, heights and volume variations, seems very promising.

**References**

45  Biemans, H., Haddeland, I., Kabat, P., Ludwig, F., Hutjes, R., Heinke, J., von Bloh, W., and Gerten, D.: Impact of reservoirs on river discharge and irrigation water supply during the 20th century, Water Resources Research, 47, 2011.

Hanasaki, N., Kanae, S., and Oki, T.: A reservoir operation scheme for global river routing models, Journal of Hydrology, 327, 22–41, 2006.

Sadki, M., Munier, S., Boone, A., and Ricci, S.: Implementation and sensitivity analysis of a Dam-Reservoir OPeration model (DROP v1. 0) over Spain, Geoscientific Model Development Discussions, pp. 1–32, 2022.

50  SciPy: $scipy.optimize.differential_evolution$, https://docs.scipy.org/doc/scipy/reference/generated/scipy.optimize.differential_evolution.html, last accessed 19 August 2022, 2022.

Shin, S., Pokhrel, Y., and Miguez-Macho, G.: High-resolution modeling of reservoir release and storage dynamics at the continental scale, Water Resources Research, 55, 787–810, 2019.

---

## Author Response (AR2)

**Response to the comments about the submitted paper *Implementation and sensitivity analysis of a Dam-Reservoir OPeration model (DROP v1.0) over Spain**

Reviewer #1 comment

This manuscript introduces the implementation and validation of a reservoir model, mostly based on the existing Hanazaki scheme, over Spain, with a focus on parameter sensitivity analysis. After reading the manuscript and carefully considering the comments from Reviewers #1 and #2 and the authors' responses, I am inclined to agree with Reviewer #1's comments, which I don't think the authors have satisfactorily addressed. In a nutshell, the insensitivity analysis is not sufficiently novel and valuable to warrant publication. There have been many existing land-river models that have adapted the Hanazaki scheme, and most have done the sensitivity analysis, more or less. The authors could shift their focus and thus elevate their study in two directions: 1) As Reviewer #1 suggested, using the model as a tool to provide some new understanding of Spain's reservoirs (and/or river systems) since there are very good observational data and now a model that works reasonably well; 2) Systematically point to the possible deficiencies in Hanazaki's scheme and feasible directions to improve based on the recent, new understanding and observational data related to reservoirs. Hanazaki's scheme has been there for so many years, and it'd be interesting (although challenging) to propose a new reservoir scheme.

Reviewer #2 comment

The authors developed a reservoir model (DROP) based on a well-established reservoir scheme introduced by Hanasaki et. al. (2006) and then did a sensitivity analysis to identify the control parameters of the reservoir scheme in Spain rivers. Overall the manuscript is well written and the narrative is easy to follow. As a new reviewer for the 2nd round review of this manuscript, I noticed that my main concerns (i.e. lack of novelty) were brought up by the previous reviewers already. However, I am satisfied by the authors' responses to the previous reviewers and think the study can be published as is.

We would like to express our gratitude for the efforts of the reviewers and for their valuable time. We think the reviewers' comments have helped us improving the manuscript and we hope that the new revised version is now  up to GMD standards.

The main concern from this review round is the lack of novelty of the study. In our opinion, the main novelty of this study is twofold.

1. First comprehensive uncertainty analysis of the Hanasaki scheme

We agree that some sensitivity tests may have been done when implementing the Hanasaki scheme into land-river models (as cited in the manuscript), but to our knowledge, no study can be found in the literature on such a comprehensive sensitivity analysis with a full exploration of statistical moments and Sobol's indices. Previous studies mainly focused on a few sensitivity criteria for one or two model parameters and what was still unclear was the sensitivity to all the parameters, as well as the interactions between them. So we think that the comprehensive sensitivity analysis using a thorough and well documented method is quite novel for this model, especially in the context of the Spanish water management practices. Also, given the importance of uncertainty quantification and sensitivity analysis in hydrological modelling (which has been more documented in the introduction of the new revised version of the manuscript), we are convinced that the results of our sensitivity

analysis could help other researchers in implementing this scheme in their modelling framework or in setting up data assimilation technics within this scheme. In the latter case, it is worth noting that qualitative understanding of parameter sensitivity is not sufficient, quantitative estimates are required to compute the error covariance matrices. Moreover, such a comprehensive sensitivity analysis represents a large amount of work that most scientists generally do not take the time to undertake (which is understandable, because it can be very time consuming and interpretation of the results is not always straightforward: it is a significant undertaking), which, we think, also increases the usefulness of our results.

We added the following text in the introduction to enhance the importance of uncertainty quantification and sensitivity analysis, and the need for a comprehensive sensitivity analysis on the reservoir scheme:

*A large number of studies can be found in the literature focusing on the sensitivity analysis of various models, in various science fields, such as machine learning (e.g., Zouhri et al., 2022) or civil engineering (e.g., Zamanian et al., 2021), etc. Pianosi et al. (2016) provided a classification of the sensitivity analysis methods used in environmental sciences and their benefits. A comprehensive sensitivity analysis, as provided by global methods such as Sobol indices, is essential for a precise understanding of a model (Saltelli et al., 2008; Saltelli, 2013). It can help to improve parameter calibration efficiency and avoid overparameterization (e.g., Shin and Jung, 2022; Tang et al., 2007a, 2007b). It is also an efficient tool to better understand the model structure (Saltelli et al., 2008), its uncertainties (e.g., Pheulpin et al., 2022) and the dominant processes under various conditions (e.g., Huang et al., 2021; Zhang et al., 2013). If understanding and quantifying uncertainties is necessary for hydrological modelling, it is also particularly critical to efficiently weight observations and model states in data assimilation technics (Liu and Gupta, 2007; Abdolghafoorian and Farhadi, 2016). Finally, for a potential extension to the global scale, in which the model parameters cannot be calibrated and validated using observations (that does not exist or are not accessible in most reservoirs of the world), a full understanding of the model sensitivity to the parameters is crucial, especially when models are used as support tools for decision-making (Herrera et al., 2022). A few sensitivity tests on the Hanasaki scheme can be found in the literature (e.g., Hanasaki et al., 2006; Shin et al., 2019), but all of them only focused on one or two parameters supposed to be the most sensitive, which may not be sufficient for a thorough understanding of the impact of parameters uncertainties (Saltelli and Annoni, 2010). To our knowledge, no comprehensive sensitivity analysis has been conducted on this scheme yet.*

2. Application of the model in standalone mode with real observations (volumes and outflows) and reconstructed inflows to exclude inflow uncertainties from the analysis

The reservoir scheme requires reservoir net inflows and water demands as inputs to estimate volume variations and outflows. In existing studies, water demands are estimated and net inflows are either modelled by a river routing scheme or estimated from gauge observations in rivers and tributaries upstream the reservoir. Reservoir abstraction is also sometimes accounted for in the reservoir water balance. But some processes are usually neglected, such as precipitation interception, direct runoff, evaporation or groundwater exchanges. This introduces a bias in the water budget and consequently increases the model uncertainties, especially when inflows are derived from land surface and river routing models (see, e.g., Vanderkelen et al., 2022). In our study, the water balance is used in a first step to derive the resultant of all these components (net inflow) from observations of reservoir volume and outflow, as described in section 3.3.2. The advantage compared to previous studies is that it removes the uncertainties related to each of these components, enabling the study of the model uncertainties themselves and the capacity of the model to reproduce the reservoir behaviour alone, without additional uncertainties or potential compensations between the components of the water budget and the model parameters.

These details have been added at the beginning of section 3.3.2 "Reconstructing inflows".

Finally, reviewer #1 suggested two directions to shift the focus of the study. The first one – providing some new understanding of Spain's reservoirs – is, in our opinion, out of the scope of the present study and could be the focus of a new article, starting for example from the calibration results presented in the supplementary material. As explained above, the objective of this study relies on the comprehensive sensitivity analysis of the DROP mode based on the Hanasaki scheme, and the region of Spain has been chosen mainly because of the reservoir dataset that is available in this country and that makes the sensitivity analysis possible. The efficiency of the DROP model over Spanish reservoirs has been explored only to assess the suitability of the model to reproduce the main behaviour of reservoirs (section 4.1).

The second direction proposed by the reviewer consists in exploring the possible deficiencies in Hanazaki scheme and feasible directions for improvement. Note that in section 5.1 of the manuscript, we present some limitations of the model that have been identified in the current study. For example, the model considers a non-realistic constant dam release for reservoirs not designed primarily for irrigation purposes. Also it is not able to account for multi-objective reservoirs or for multi-reservoir systems. Release rules are also too simple to represent complex socio-economic and political factors, which are furthermore different in each country.

Thus, in summary of our response: we have added considerable discussion in the Introduction in order to document the importance of doing a parameter sensitivity analysis using a sophisticated and statistically-based method, which has never been done with the Hanasaki scheme. We have also added text to highlight the additional novelty of this study in section 3.2.2. Finally, we feel that the goal of this study is not to focus on improved understanding of Spanish reservoirs per say: we have chosen this region owing to the availability of data (since such data needed for a robust evaluation of such a model is not always public domain in some countries such as, for example, France). We address the second proposition of this reviewer in Section 5.1 as explained above. We sincerely hope that the additional text (including new references) have added the suggested focus and address the questions proposed by reviewer #1.

Abdolghafoorian, A., & Farhadi, L. (2016). Uncertainty quantification in land surface hydrologic modeling: Toward an integrated variational data assimilation framework. IEEE Journal of Selected Topics in Applied Earth Observations and Remote Sensing, 9(6), 2628-2637.

Cibin, R., Sudheer, K. P., & Chaubey, I. (2010). Sensitivity and identifiability of stream flow generation parameters of the SWAT model. Hydrological Processes: An International Journal, 24(9), 1133-1148.

Herrera, P. A., Marazuela, M. A., & Hofmann, T. (2022). Parameter estimation and uncertainty analysis in hydrological modeling. Wiley Interdisciplinary Reviews: Water, 9(1), e1569.

Huang, C., Tong, J., & Ye, M. (2021). Global sensitivity analysis for a prediction model of soil solute transfer into surface runoff. Journal of Hydrology, 599, 126342.

Liu, Y., & Gupta, H. V. (2007). Uncertainty in hydrologic modeling: Toward an integrated data assimilation framework. Water resources research, 43(7).

Pheulpin, L., Bertrand, N., & Bacchi, V. (2022). Uncertainty quantification and global sensitivity analysis with dependent inputs parameters: Application to a basic 2D-hydraulic model. LHB, 108(1), 2015265.

Pianosi, F., Beven, K., Freer, J., Hall, J. W., Rougier, J., Stephenson, D. B., & Wagener, T. (2016). Sensitivity analysis of environmental models: A systematic review with practical workflow. Environmental Modelling & Software, 79, 214-232.

Saltelli, A., and Annoni P., 2010, How to avoid a perfunctory sensitivity analysis, Environmental Modeling and Software, 25 1508-1517.

Saltelli, A., 2013, The cautious modeller: craftsmanship without wizardry. Preface to 'Analyse de sensibilité et exploration de modèles. Applications aux modèles environnementaux', Robert Faivre, Bertrand Iooss, Stéphanie Mahévas, David Makowski, and Hervé Monod Editors, Edition QUAE.

Shin, M. J., & Jung, Y. (2022). Using a global sensitivity analysis to estimate the appropriate length of calibration period in the presence of high hydrological model uncertainty. Journal of Hydrology, 607, 127546.

Tang, Y., Reed, P., Van Werkhoven, K., & Wagener, T. (2007a). Advancing the identification and evaluation of distributed rainfall-runoff models using global sensitivity analysis. Water Resources Research, 43(6).

Tang, Y., Reed, P., Wagener, T., & Van Werkhoven, K. (2007b). Comparing sensitivity analysis methods to advance lumped watershed model identification and evaluation. Hydrology and Earth System Sciences, 11(2), 793-817.

Vanderkelen, I., Gharari, S., Mizukami, N., Clark, M. P., Lawrence, D. M., Swenson, S., … Thiery, W. (2022). Evaluating a reservoir parametrization in the vector-based global routing model mizuRoute (v2.0.1) for Earth system model coupling. Geoscientific Model Development, 15(10), 4163–4192. https://doi.org/10.5194/gmd-15-4163-2022

Zamanian, S., Hur, J., & Shafieezadeh, A. (2021). Significant variables for leakage and collapse of buried concrete sewer pipes: A global sensitivity analysis via Bayesian additive regression trees and Sobol'indices. Structure and infrastructure engineering, 17(5), 676-688.

Zhang, C., Chu, J., & Fu, G. (2013). Sobol''s sensitivity analysis for a distributed hydrological model of Yichun River Basin, China. Journal of hydrology, 480, 58-68.

Zouhri, W., Homri, L., & Dantan, J. Y. (2022). Handling the impact of feature uncertainties on SVM: a robust approach based on Sobol sensitivity analysis. Expert Systems with Applications, 189, 115691.